# Chaotic dynamics and the role of covariance inflation for reduced rank Kalman filters with model error

Colin Grudzien[1], Alberto Carrassi[1], and Marc Bocquet[2]

[1]Nansen Environmental and Remote Sensing Center, Bergen, Norway
[2]CEREA, joint laboratory École des Ponts ParisTech and EDF R&D, Université Paris-Est, Champs-sur-Marne, France

*Correspondence to:* Colin Grudzien, Colin.Grudzien@nersc.no

**Abstract.** The ensemble Kalman filter and its variants have shown to be robust for data assimilation in high dimensional geophysical models, with localization, using ensembles of extremely small size relative to the model dimension. A reduced rank representation of the estimated covariance, however, leaves a large dimensional complementary subspace unfiltered. Utilizing the dynamical properties of the filtration for the backward Lyapunov vectors, this paper explores a previously unexplained mechanism, providing a novel theoretical interpretation for the role of covariance inflation in ensemble-based Kalman filters. Our derivation of the forecast error evolution describes the dynamic upwelling of the unfiltered error from outside of the span of the anomalies into the filtered subspace. Analytical results for linear systems explicitly describe the mechanism for the upwelling, and the associated recursive Riccati equation for the forecast error, while nonlinear approximations are explored numerically.

## 1 Introduction

It is well understood that in chaotic physical systems, dynamical instability is among the leading drivers of forecast uncertainty (Toth and Kalnay, 1997; Trevisan and Palatella, 2011a; Vannitsem, 2017). Recent mathematical and numerical results have, furthermore, established a rigorous framework for understanding the relationship between dynamical instability, in terms of the non-negative Lyapunov exponents, and the asymptotic properties of the uncertainty in ensemble-based data assimilation techniques: in perfect models, with weakly-nonlinear error growth, the anomalies of ensemble Kalman filters project strongly on the span of the unstable-neutral backward Lyapunov vectors (Carrassi et al., 2009; Ng et al., 2011; González-Tokman and Hunt, 2013; Bocquet and Carrassi, 2017), and that the divergence of the ensemble Kalman filter depends significantly upon whether error in this space is sufficiently observed and corrected.

Inspired by the Assimilation in the Unstable Subspace (AUS) methodology of Anna Trevisan and her collaborators (Trevisan and Uboldi, 2004; Carrassi et al., 2007, 2008; Trevisan et al., 2010; Trevisan and Palatella, 2011b; Palatella et al., 2013; Palatella and Trevisan, 2015), recent mathematical results have rigorously validated the underlying hypothesis of AUS: for perfect, linear models, the uncertainty of the Kalman filter asymptotically collapses to the span of the backward Lyapunov vectors with non-negative exponents (Gurumoorthy et al., 2017). Furthermore, if a reduced rank filter has an estimated covariance initialized only in these modes, and the unstable-neutral subspace is uniformly, completely observed, the reduced rank filter be-

comes asymptotically equivalent to the optimal Kalman filter (Bocquet et al., 2017). This phenomenon has, furthermore, been generalized as a necessary and sufficient criterion for the exponential stability of continuous time filters, in perfect models, in terms of the detectability of the unstable-neutral subspace (Frank and Zhuk, 2017).

The above mathematical studies demonstrate how the stable dynamics in perfect models dissipate forecast errors, in sequential filters, such that a reduced rank representation of the error covariance matrix in the unstable-neutral subspace alone suffices to control error growth. This behavior, similarly understood in the smoothing problem (Pires et al., 1996; Trevisan et al., 2010), is now also mathematically verified for the linear Kalman smoother and its nonlinear ensemble formulation is shown numerically to exhibit the same behavior in a weakly-nonlinear regime for error dynamics (Bocquet and Carrassi, 2017). With these results, AUS provides a robust theoretical framework for interpreting the behavior of the ensemble Kalman filter in terms of the model dynamics. However, this framework has, for the most part, been limited to understanding the ensemble Kalman filter in models without errors.

The sources of model error are varied and a common simplifying assumption in data assimilation is that it takes the form of additive, Gaussian noise that is white in time. The work of Grudzien et al. (2018) recently extended the theoretical framework of AUS, so far established for perfect models, to the presence of additive model errors with additional qualifications. That study introduced novel bounds on the Kalman filter's asymptotic forecast uncertainty, and a necessary criterion for filter stability, as an inverse relationship between the model's dynamical instabilities and the relative precision of observations. Particularly, in stationary dynamics and the absence of corrections to forecast errors in the stable modes, that work demonstrated that the model dynamics alone are once again sufficient to uniformly bound the errors in the span of the stable backward Lyapunov vectors.

However, the uniform bound may be impractically large due to the excitation of model errors by the transient instabilities in stable directions. While uncertainty is asymptotically dissipated by the stable dynamics, the reintroduction of uncertainty from model error significantly differentiates models with additive errors. Newly injected errors are subject to the growth rates of the local (in time) Lyapunov exponents, and stable Lyapunov exponents of sufficiently high variance may experience transient periods of growth. Therefore, strategies for representing the forecast error with a low rank ensemble must be adapted for imperfect models to account for a residual error in the span of the stable, backward Lyapunov vectors which never vanishes and, moreover, may go through transient periods of growth. As a consequence, confining the error description within a reduced rank Kalman filter to only the unstable-neutral subspace does not suffice when model error is present and suggests that one must include additional, asymptotically stable, modes.

In this current work we show, furthermore, that such an increase of the ensemble span does not automatically render the filter optimal: one may also need to account for the injection of error from unfiltered directions into the ensemble span. In particular, when an ensemble-based Kalman gain is used to correct the forecast errors, the dynamics induce error propagation which transmits uncertainty from the uncorrected, complementary subspace into the ensemble span. In this study, the propagation of error in the linear Kalman filter, written in a basis of backward Lyapunov vectors, will reveal the leading order evolution of the unfiltered uncertainty. Although the evolution is derived for linear models, the mechanism for error propagation can be considered a generic feature of ensemble Kalman filters. Under the condition that error evolution is weakly-nonlinear, the

ensemble span will align with the span of the leading backward Lyapunov vectors — therefore the error decomposition in the basis of backward Lyapunov vectors will be valid for the ensemble Kalman filter.

Similar to how we view AUS as a theoretical framework for understanding the properties of ensemble-based covariances in the presence of chaotic dynamics (and in the absence of model error), this work aims to be used as a theoretical explanation for the empirically observed properties of ensemble-based covariances in the presence of chaotic dynamics and additive model errors, providing a theoretical motivation for the role of covariance inflation in preventing filter divergence. We demonstrate that even when issues of sampling error, truncation errors due to nonlinearity, and misspecification of model and observation error distributions are all excluded, there is an intrinsic deficiency of the standard reduced rank error covariance recursion that leads to systematic underestimation of the forecast errors in the ensemble span. While we believe this provides a new theoretical explanation for the role of covariance inflation in the ensemble Kalman filter, we also discuss possible strategies to obviate inflation with less ad hoc methods that take into account the evolution of unfiltered errors more directly.

This paper is structured as follows: section 2.1 concerns essential results from the theory of Lyapunov vectors which are used throughout; sections 2.2 and 2.3 describe the basic framework for the Kalman filter, and will motivate our subsequent results; section 3 contains our main analytical result, the derivation of the exact forecast error under a reduced rank filter in a basis of backward Lyapunov vectors; section 4 will use numerics to qualitatively explore the forecast error of the reduced rank filter, and its approximation in nonlinear models. Implications of the results in this work are discussed in section 5, with an emphasis on future directions of research and their challenges. Final conclusions are drawn in section 6.

## 2  Preliminaries

We begin by introducing our notation and the problem formulation, with definitions in bold. There is inconsistent use of the terminology for Lyapunov vectors in the literature, and so we choose to use the nomenclature of Kuptsov and Parlitz (2012) for its accessibility and self-consistency.

### 2.1  Lyapunov vectors

Throughout the entire text, the conventional notation $k = 0, 1, 2, \ldots$ is adopted to indicate that the quantity is defined at time $t_k$. Let $\mathbf{z}_{k-1} \in \mathbb{R}^n$ be an arbitrary vector, the matrix propagator of the forward model from $t_{k-1}$ to $t_k$ is given by $\mathbf{M}_k$, such that $\mathbf{z}_k = \mathbf{M}_k \mathbf{z}_{k-1}$. We denote the operator taking the system state from an arbitrary time $t_l$ to $t_k$ as $\mathbf{M}_{k:l} \triangleq \mathbf{M}_k \mathbf{M}_{k-1} \ldots \mathbf{M}_{l+1}$, with the symbol $\triangleq$ used to signify that the expression is a definition. We denote $\mathbf{M}_{k:k} \triangleq \mathbf{I}_n$, where $\mathbf{I}_n$ is the identity matrix (of size $n \times n$ in this case). At all times we assume $\mathbf{M}_k$ to be non-singular and to be uniformly bounded in $k$.

Although much of the derivations that follow are done for linear dynamics, we are ultimately concerned with nonlinear systems — therefore, we will assume that Oseledec's theorem holds, even for linear model propagators. In general, this is a non-trivial assumption, but one which can be considered generic for the tangent-linear model of a wide class of nonlinear systems, due to the Multiplicative Ergodic Theorem (**MET**): with probability one, Oseledec's theorem holds, the Lyapunov exponents are well defined and the values of the Lyapunov exponents are independent of the initial condition (Barreira and

Pesin, 2002, see their Theorems 2.1.1 and 2.1.2 for a full statement and proof). A more general version of the MET, and its interpretation for several physical systems, is provided by Froyland et al. (2013) in their Theorem 1.1 and example 1.2.

We order the Lyapunov exponents

$$\lambda_1 \geq \cdots \geq \lambda_{n_0} \geq 0 > \lambda_{n_0+1} \geq \cdots \geq \lambda_n, \tag{1}$$

such that the unstable-neutral subspace is of dimension $n_0$ and the model state dimension is $n$. Note that we do not assume that the Lyapunov exponents are distinct.

Oseledec's theorem decomposes the (tangent-linear) model space into a direct sum of time-varying, covariant Oseledec spaces, referred to as an Oseledec splitting or decomposition. At times, we will refer to the covariant Oseledec spaces, as well as to the covariant, and to the forward Lyapunov vectors. These discussions will provide a deeper interpretation of our results

for those familiar with these technical points. However, these discussions are not crucial to the understanding of our results, and we therefore limit the use of formal definitions to the backward Lyapunov vectors. For a more formal discussion of the Oseledec spaces, constructions for Lyapunov vectors and related results for the full rank Kalman filter, see Grudzien et al. (2018); for a survey on the mathematical and numerical construction of Lyapunov vectors, see Kuptsov and Parlitz (2012); for a discussion of general Oseledec splitting, and a comparison of methods for its computation, see Froyland et al. (2013).

The backward Lyapunov vectors can be defined by a choice of an orthonormal eigenbasis for the far-past operator, and/or by recursive QR factorizations of the (tangent-linear) model propagator (Kuptsov and Parlitz, 2012). Throughout the text, we utilize the invariance of the backward Lyapunov vectors under the recursive QR algorithm.

**Definition 1.** *Define the matrix $\mathbf{E}_k$ to be the orthogonal matrix at time $k$ whose $i$-th column is the $i$-th **backward Lyapunov vector (BLV)**, corresponding to the Lyapunov exponent $\lambda_i$.*

**Lemma 1.** *There is an $n \times n$ upper triangular matrix $\mathbf{U}_k$, such that the (tangent-linear) model propagator satisfies*

$$\mathbf{M}_k = \mathbf{E}_k \mathbf{U}_k \mathbf{E}_{k-1}^{\mathrm{T}}. \tag{2}$$

*Define the product of matrices,*

$$\mathbf{U}_{k:l} \triangleq \mathbf{U}_k \cdots \mathbf{U}_l, \tag{3}$$

*the $i$-th Lyapunov exponent is equal to the limit*

$$\lambda_i = \lim_{k \to \infty} \frac{1}{k-l} \log\left(|U_{k:l}^{ii}|\right), \tag{4}$$

*where $U_{k:l}^{ii}$ is the $i$-th diagonal element of the matrix $\mathbf{U}_{k:l}$. The local Lyapunov exponents are defined by $\log\left(|U_k^{ii}|\right)$.*

*Proof.* Equation (2) follows from Eq. (31) of Kuptsov and Parlitz (2012) and is a consequence of the invariance of the BLVs under the recursive QR decomposition (Grudzien et al., 2018). Computing Lyapunov exponents via recursive QR factorizations as in Eq. (4) is the standard method, described by e.g., Shimada and Nagashima (1979), Benettin et al. (1980) and Ershov and

Potapov (1998).  □

The decomposition in Eq. (2) represents a change of basis of the model space into the upper triangular dynamics of the moving frame of BLVs, defining a basis for the backward Lyapunov filtration (Legras and Vautard, 1996). In particular, $\mathbf{E}_{k-1}^{\mathrm{T}}$ takes the model state into the orthogonal projection coefficients in the basis of the BLVs at time $k-1$. We will denote the projection coefficients of an arbitrary vector $\mathbf{z}_k$ into a basis of BLVs with a "hat", i.e. $\mathbf{E}_k^{\mathrm{T}} \mathbf{z}_k \triangleq \widehat{\mathbf{z}}_k$. Using the orthogonality of the matrix $\mathbf{E}_k$, the invariant dynamics in the BLVs is written

$$\widehat{\mathbf{z}}_k = \mathbf{U}_k \widehat{\mathbf{z}}_{k-1} \quad \Leftrightarrow \quad \mathbf{z}_k = \mathbf{M}_k \mathbf{z}_{k-1}. \tag{5}$$

The operator $\mathbf{U}_k$ thus describes the invariant, upper triangular dynamics, transferring the model state into its forward representation in the BLVs at time $k$.

## 2.2 The Kalman filter

We seek to estimate the distribution of a Gaussian random vector $\mathbf{x}_k \in \mathbb{R}^n$ evolved via a linear Markov model with additive white noise,

$$\mathbf{x}_k = \mathbf{M}_k \mathbf{x}_{k-1} + \mathbf{w}_k, \tag{6}$$

and with observations $\mathbf{y}_k \in \mathbb{R}^d$ given in the form

$$\mathbf{y}_k = \mathbf{H}_k \mathbf{x}_k + \mathbf{v}_k. \tag{7}$$

The forecast mean, $\mathbf{x}_k^{\mathrm{b}}$, is propagated from the last posterior mean, $\mathbf{x}_{k-1}^{\mathrm{a}}$ by the deterministic component of Eq. 6, i.e.,

$$\mathbf{x}_k^{\mathrm{b}} = \mathbf{M}_k \mathbf{x}_{k-1}^{\mathrm{a}}. \tag{8}$$

The model variables and observation vectors are related via the linear observation operator $\mathbf{H}_k : \mathbb{R}^n \mapsto \mathbb{R}^d$. Model and observation noise, $\mathbf{w}_k$ and $\mathbf{v}_k$, are assumed mutually independent, unbiased, Gaussian white sequences such that

$$\mathbb{E}[\mathbf{v}_k \mathbf{v}_l^{\mathrm{T}}] = \delta_{k,l} \mathbf{R}_k \quad \text{and} \quad \mathbb{E}[\mathbf{w}_k \mathbf{w}_l^{\mathrm{T}}] = \delta_{k,l} \mathbf{Q}_k, \tag{9}$$

where $\mathbb{E}$ is the expectation, $\mathbf{R}_k \in \mathbb{R}^{d \times d}$ is the observation error covariance matrix at time $t_k$, and $\mathbf{Q}_k \in \mathbb{R}^{n \times n}$ stands for the model error covariance matrix. The error covariance matrix $\mathbf{R}_k$ can be assumed invertible without losing generality. To avoid pathologies, we assume that the model and the observation error covariance matrices are uniformly bounded. For $1 \leq t < s \leq n$, and given a matrix $\mathbf{A} \in \mathbb{R}^{n \times n}$, we define $\mathbf{A}^{t:s} \in \mathbb{R}^{n \times (s-t+1)}$ to be the matrix composed (inclusively) of columns $s$ through $t$ of $\mathbf{A}$.

**Definition 2.** *The **forecast error** is defined as the difference of the mean state estimated by the filter and the unknown random state, i.e.,*

$$\boldsymbol{\epsilon}_k \triangleq \mathbf{x}_k^{\mathrm{b}} - \mathbf{x}_k. \tag{10}$$

*The **innovation** is the measured difference between the forecast in the observation space and the observation,*

$$\boldsymbol{\delta}_k \triangleq \mathbf{y}_k - \mathbf{H}_k \mathbf{x}_k^{\mathrm{b}} = \mathbf{v}_k - \mathbf{H}_k \boldsymbol{\epsilon}_k. \tag{11}$$

*We define the **exact forecast error covariance** at time $k$ to be*

$$\mathbf{B}_k \triangleq \mathbb{E}\left[\boldsymbol{\epsilon}_k \boldsymbol{\epsilon}_k^{\mathrm{T}}\right]. \tag{12}$$

*On the other hand, suppose some filter, yet to be identified, is used to estimate the forecast mean and error covariance — the **estimated forecast error covariance** will be denoted $\mathbf{P}_k$, defined according to the chosen estimation algorithm.*

Suppose that $\mathbf{K}_k \in \mathbb{R}^{n \times d}$ is some estimator which takes the forecast state to the analysis state. In the case of the theoretical Kalman filter, where the exact forecast error covariances are computed $\mathbf{P}_k \equiv \mathbf{B}_k$, the gain $\mathbf{K}_k$ will be defined

$$\begin{aligned}
\mathbf{K}_k &\triangleq \mathbf{P}_k \mathbf{H}_k^{\mathrm{T}} \left(\mathbf{H}_k \mathbf{P}_k \mathbf{H}_k^{\mathrm{T}} + \mathbf{R}_k\right)^{-1} \\
&= \mathbf{B}_k \mathbf{H}_k^{\mathrm{T}} \left(\mathbf{H}_k \mathbf{B}_k \mathbf{H}_k^{\mathrm{T}} + \mathbf{R}_k\right)^{-1}.
\end{aligned} \tag{13}$$

In this text, we will vary the choice of the analysis update operator $\mathbf{K}_k$, but the functional form of the recursion for the analysis update of the mean will be unchanged and defined as

$$\begin{aligned}
\mathbf{x}_k^{\mathrm{a}} &\triangleq \mathbf{x}_k^{\mathrm{b}} + \mathbf{K}_k \left(\mathbf{y}_k - \mathbf{H}_k \mathbf{x}_k^{\mathrm{b}}\right) \\
&= \mathbf{x}_k^{\mathrm{b}} - \mathbf{K}_k \mathbf{H}_k \boldsymbol{\epsilon}_k + \mathbf{K}_k \mathbf{v}_k.
\end{aligned} \tag{14}$$

Therefore, for any estimator, the forecast mean can be derived recursively from Eq. (8) and Eq. (14) as

$$\mathbf{x}_{k+1}^{\mathrm{b}} \triangleq \mathbf{M}_{k+1} \left(\mathbf{x}_k^{\mathrm{b}} - \mathbf{K}_k \mathbf{H}_k \boldsymbol{\epsilon}_k + \mathbf{K}_k \mathbf{v}_k\right) \tag{15}$$

where $\mathbf{K}_k$ is some choice for the gain. The recursion on the forecast error can be derived equal to

$$\boldsymbol{\epsilon}_{k+1} \triangleq \mathbf{M}_{k+1} \left[\left(\mathbf{I}_n - \mathbf{K}_k \mathbf{H}_k\right) \boldsymbol{\epsilon}_k + \mathbf{K}_k \mathbf{v}_k\right] - \mathbf{w}_{k+1}, \tag{16}$$

though $\boldsymbol{\epsilon}_k, \mathbf{v}_k$ and $\mathbf{w}_{k+1}$ are assumed to be unknown.

## 2.3  Rank deficiency in the Kalman filter

In a linear model, with known Gaussian observation and model error distributions, the estimated error covariances of the KF are exact: the posterior error distribution for the state is Gaussian, and the KF completely describes the Bayesian posterior through its recursive equations for the estimated mean and covariance. However, it is often the case that the recursion for the posterior error distribution is approximated with a reduced rank surrogate in which the estimated covariance, $\mathbf{P}_k$, and resulting exact

error covariance, $\mathbf{B}_k$, may not be equal (Chandrasekar et al., 2008). This mis-match can lead to systematic underestimation of the forecast error and filter divergence.

Nonetheless, it is possible in an ideal setting to analytically describe the error statistics of a reduced rank Kalman filter — to illustrate this, assume that we have a linear model with known Gaussian error distributions. Suppose we apply the analysis update in a reduced rank set of BLVs, as has been done in EKF-AUS (Trevisan and Palatella, 2011b). Suppose, furthermore, the exact error covariance, $\mathbf{B}_k$, is known. Then the gain

$$\mathbf{K}_k \triangleq \mathbf{E}_k^{1:n_0} \left( \mathbf{E}_k^{1:n_0} \right)^{\mathrm{T}} \mathbf{B}_k \mathbf{E}_k^{1:n_0} \left( \mathbf{E}_k^{1:n_0} \right)^{\mathrm{T}} \mathbf{H}_k^{\mathrm{T}} \times$$

$$\left[ \mathbf{H}_k \mathbf{E}_k^{1:n_0} \left( \mathbf{E}_k^{1:n_0} \right)^{\mathrm{T}} \mathbf{B}_k \mathbf{E}_k^{1:n_0} \left( \mathbf{E}_k^{1:n_0} \right)^{\mathrm{T}} \mathbf{H}_k^{\mathrm{T}} + \mathbf{R}_k \right]^{-1} \tag{17}$$

yields the exact Kalman estimator with respect to a subset of the anomaly variables, defined by the span of the leading $n_0$ BLVs. We may use Eq. (16) to derive the analytical recursion for the forecast error covariance, $\mathbf{B}_{k+1}$, under the reduced rank gain in Eq. (17). The **rank deficiency (or reduced rank)** is defined by the restriction of the Kalman estimator to a low dimensional

subspace. Note that, although the estimator is restricted to the span of $\mathbf{E}_k^{1:n_0}$, the observation operator is still applied to the full state vector, and thus the analysis does not equal the restriction of the Bayesian update to the leading $n_0$ BLVs. We recover the restricted Bayesian update using the estimator in Eq. (17) precisely when $\mathbf{H}_k \mathbf{E}_k^{n_0+1:n} \equiv \mathbf{0}_{d \times (n-n_0)}$.

The significance of deriving an analytical recursion for the forecast error under the reduced rank estimator in Eq. (17) is as follows. The analysis operator in Eq. (17) is characteristic of the typical gain for the ensemble Kalman filter (**EnKF**) in

large, geophysical models: the ensemble-based gain applies its update with respect to the subspace defined by the span of the ensemble of anomalies, which is usually of reduced rank and aligns with the span of the leading BLVs (Ng et al., 2011; Bocquet and Carrassi, 2017). The standard EnKF can, therefore, be considered a Monte Carlo estimate of the error statistics resulting from a rank deficient Kalman estimator as in Eq. (17). This is the motivation of section 3, where we will define a reduced rank gain which operates within the span of an arbitrary number of the leading BLVs and derive the resulting exact forecast error

covariance.

## 3   Filtering in the backward Lyapunov basis vectors

Consider the forecast error recursion for the linear KF in Eq. (16). As we are motivated by ensemble covariances, suppose $\mathbf{K}_k$ is defined as a reduced rank gain which corrects only the leading $r$ BLVs, with $r < n$. The subspace defined by the span of the anomalies defines a subspace of "filtered variables" where we perform our analysis. The "unfiltered subspace" is uniquely

defined (up to the inner product) as the orthogonal complement to the filtered space, i.e., the subspace in which the reduced rank Kalman estimator makes no correction.

**Definition 3.** *For each $k$, we define the **filtered subspace** by the column span of the vectors $\mathbf{E}_k^{\mathrm{f}} \triangleq \mathbf{E}_k^{1:r}$ and the **unfiltered subspace** by the column span of the vectors $\mathbf{E}_k^{\mathrm{u}} \triangleq \mathbf{E}_k^{r+1:n}$. The projection coefficients of a vector $\mathbf{z} \in \mathbb{R}^n$ into the filtered and unfiltered subspace will be denoted $\widehat{\mathbf{z}}^{\mathrm{f}} \triangleq \left( \mathbf{E}_k^{\mathrm{f}} \right)^{\mathrm{T}} \mathbf{z}$ and $\widehat{\mathbf{z}}^{\mathrm{u}} \triangleq \left( \mathbf{E}_k^{\mathrm{u}} \right)^{\mathrm{T}} \mathbf{z}$, respectively.*

We thus decompose the forecast error into its orthogonal projections in the filtered and unfiltered subspaces as

$$\epsilon_k \triangleq \mathbf{E}_k^{\mathrm{f}} \widehat{\epsilon}_k^{\mathrm{f}} + \mathbf{E}_k^{\mathrm{u}} \widehat{\epsilon}_k^{\mathrm{u}}. \tag{18}$$

For $r = n$, define $\mathbf{E}_k^{\mathrm{f}} \triangleq \mathbf{E}_k$ and $\mathbf{E}_k^{\mathrm{u}} \triangleq \mathbf{0}_n$ such that $\widehat{\boldsymbol{\epsilon}}_k^{\mathrm{f}}$ is the full error written in an orthogonal change of basis — this case will only be referred to for comparison.

For $i, j \in \{\mathrm{f}, \mathrm{u}\}$, we write the sub-covariances in the basis defined by $\mathbf{E}_k$ as

$$\widehat{\mathbf{B}}_k^{ij} \triangleq \mathbb{E}\left[\widehat{\boldsymbol{\epsilon}}_k^i \left(\widehat{\boldsymbol{\epsilon}}_k^j\right)^{\mathrm{T}}\right]. \tag{19}$$

such that the exact forecast error covariance is given

$$\mathbf{B}_k \equiv \mathbf{E}_k \begin{pmatrix} \widehat{\mathbf{B}}_k^{\mathrm{ff}} & \widehat{\mathbf{B}}_k^{\mathrm{fu}} \\ \widehat{\mathbf{B}}_k^{\mathrm{uf}} & \widehat{\mathbf{B}}_k^{\mathrm{uu}} \end{pmatrix} \mathbf{E}_k^{\mathrm{T}}, \tag{20}$$

where $\widehat{\mathbf{B}}_k^{\mathrm{ff}}$ and $\widehat{\mathbf{B}}_k^{\mathrm{uu}}$ are symmetric matrices, and $\widehat{\mathbf{B}}_k^{\mathrm{fu}} = \left(\widehat{\mathbf{B}}_k^{\mathrm{uf}}\right)^{\mathrm{T}}$. We similarly express $\mathbf{U}_k$ as a block matrix,

$$\mathbf{U}_k \triangleq \begin{pmatrix} \mathbf{U}_k^{\mathrm{ff}} & \mathbf{U}_k^{\mathrm{fu}} \\ \mathbf{0}_{(n-r)\times r} & \mathbf{U}_k^{\mathrm{uu}} \end{pmatrix}. \tag{21}$$

For an arbitrary rank filtered subspace, the reduced rank gain $\mathbf{K}_k$ correcting the span of $\mathbf{E}_k^{\mathrm{f}}$ is defined by

$\mathbf{K}_k \triangleq \mathbf{E}_k^{\mathrm{f}} \widehat{\mathbf{K}}_k,$

$$\widehat{\mathbf{K}}_k \triangleq \mathbf{B}_k^{\mathrm{ff}} \left(\mathbf{E}_k^{\mathrm{f}}\right)^{\mathrm{T}} \mathbf{H}_k^{\mathrm{T}} \left[\mathbf{H}_k \mathbf{E}_k^{\mathrm{f}} \mathbf{B}_k^{\mathrm{ff}} \left(\mathbf{E}_k^{\mathrm{f}}\right)^{\mathrm{T}} \mathbf{H}_k^{\mathrm{T}} + \mathbf{R}_k\right]^{-1}, \tag{22}$$

where $\widehat{\mathbf{K}}_k$ represents the projection coefficients of the reduced rank gain into the filtered variables.

For every $k \geq 1$, we decompose the model error covariance into the basis of filtered and unfiltered BLVs as

$$\mathbf{Q}_k \triangleq \mathbf{E}_k \begin{pmatrix} \widehat{\mathbf{Q}}_k^{\mathrm{ff}} & \widehat{\mathbf{Q}}_k^{\mathrm{fu}} \\ \widehat{\mathbf{Q}}_k^{\mathrm{uf}} & \widehat{\mathbf{Q}}_k^{\mathrm{uu}} \end{pmatrix} \mathbf{E}_k^{\mathrm{T}} \tag{23}$$

where $\widehat{\mathbf{Q}}_k^{\mathrm{ff}}$ and $\widehat{\mathbf{Q}}_k^{\mathrm{uu}}$ are symmetric matrices, and $\widehat{\mathbf{Q}}_k^{\mathrm{fu}} = \left(\widehat{\mathbf{Q}}_k^{\mathrm{uf}}\right)^{\mathrm{T}}$.

With the above notation, and using Eq. (2), the evolution of the forecast error under the reduced rank gain is derived from Eq. (16) as

$$\begin{aligned}
\boldsymbol{\epsilon}_{k+1} &= \mathbf{M}_{k+1}\left(\mathbf{I}_n - \mathbf{E}_k^{\mathrm{f}} \widehat{\mathbf{K}}_k \mathbf{H}_k\right)\boldsymbol{\epsilon}_k + \mathbf{M}_{k+1}\mathbf{E}_k^{\mathrm{f}}\widehat{\mathbf{K}}_k \mathbf{v}_k - \mathbf{w}_{k+1} \\
&= \left(\mathbf{E}_{k+1}\mathbf{U}_{k+1}\mathbf{E}_k^{\mathrm{T}} - \mathbf{E}_{k+1}\mathbf{U}_{k+1}\mathbf{I}_{n\times r}\widehat{\mathbf{K}}_k \mathbf{H}_k\right)\boldsymbol{\epsilon}_k + \mathbf{E}_{k+1}\mathbf{U}_{k+1}\mathbf{I}_{n\times r}\widehat{\mathbf{K}}_k \mathbf{v}_k - \mathbf{w}_{k+1}.
\end{aligned} \tag{24}$$

Equation (24) describes the evolution of the forecast error with respect to the sub-optimal filter, and suggests, as in Eq. (5), how we may write the error evolution into the upper triangular dynamics in the moving frame of BLVs. Computing the evolution of $\widehat{\boldsymbol{\epsilon}}_k^{\mathrm{f}}$ and $\widehat{\boldsymbol{\epsilon}}_k^{\mathrm{u}}$ under the forecast-analysis update cycle in Eq. (24), we will derive the exact recursion for $\widehat{\mathbf{B}}_k^{\mathrm{ff}}$. This will describe the exact forecast uncertainty in the filtered subspace under a gain which operates in the span of the leading $r$ BLVs.

## 3.1 Evolution of unfiltered error

We begin by deriving the evolution of error in the unfiltered subspace, by verifying that it evolves according to the free evolution. Notice first the following relation,

$$\left(\mathbf{E}_{k+1}^{\mathrm{u}}\right)^{\mathrm{T}} \mathbf{E}_{k+1} \mathbf{U}_{k+1} \mathbf{I}_{n \times r} = \mathbf{0}_{(n-r) \times r}, \tag{25}$$

due to the fact that $\mathbf{E}_{k+1}$ is an orthogonal matrix and, therefore, that the above product is equal to the lower left block of $\mathbf{U}_{k+1}$, which is upper triangular. With substitution of Eq. (18) into in Eq. (24) for $\boldsymbol{\epsilon}_k$, multiplying on the left by $\left(\mathbf{E}_k^{\mathrm{u}}\right)^{\mathrm{T}}$ to move into the unfiltered subspace, and by utilizing Eq. (25) to cancel the error in the filtered space, we find

$$\widehat{\boldsymbol{\epsilon}}_{k+1}^{\mathrm{u}} = \left(\mathbf{E}_{k+1}^{\mathrm{u}}\right)^{\mathrm{T}} \mathbf{E}_{k+1} \mathbf{U}_{k+1} \mathbf{E}_k^{\mathrm{T}} \left(\mathbf{E}_k^{\mathrm{f}} \widehat{\boldsymbol{\epsilon}}_k^{\mathrm{f}} + \mathbf{E}_k^{\mathrm{u}} \widehat{\boldsymbol{\epsilon}}_k^{\mathrm{u}}\right) - \left(\mathbf{E}_{k+1}^{\mathrm{u}}\right)^{\mathrm{T}} \mathbf{w}_{k+1} \tag{26}$$

$$= \mathbf{U}_{k+1}^{\mathrm{uu}} \widehat{\boldsymbol{\epsilon}}_k^{\mathrm{u}} - \widehat{\mathbf{w}}_{k+1}^{\mathrm{u}}. \tag{27}$$

Equation (27) demonstrates that the evolution of the error in the unfiltered subspace follows exactly the free forecast evolution. The covariance of unfiltered error at time $k$ can be computed from Eq. (27) as

$$\widehat{\mathbf{B}}_k^{\mathrm{uu}} = \mathbf{U}_{k:0}^{\mathrm{uu}} \widehat{\mathbf{B}}_0^{\mathrm{uu}} \left(\mathbf{U}_{k:0}^{\mathrm{uu}}\right)^{\mathrm{T}} + \sum_{l=1}^{k} \mathbf{U}_{k:l}^{\mathrm{uu}} \widehat{\mathbf{Q}}_l^{\mathrm{uu}} \left(\mathbf{U}_{k:l}^{\mathrm{uu}}\right)^{\mathrm{T}}. \tag{28}$$

The initial uncertainty in the unfiltered subspace evolves as $\mathbf{U}_{k:0}^{\mathrm{uu}} \widehat{\mathbf{B}}_0^{\mathrm{uu}} \left(\mathbf{U}_{k:0}^{\mathrm{uu}}\right)^{\mathrm{T}}$ and thus, when $r > n_0$, vanishes exponentially. This implies that asymptotic unfiltered error is independent of the initialization, similar to the results of Bocquet et al. (2017). The remaining sum in Eq. (28) represents the contribution to the current forecast uncertainty from the model error at all times after initialization, propagated under the upper triangular evolution in the BLVs. Therefore, while the initial error is forgotten, the asymptotic error in the reduced rank filter here explicitly depends on the dimension of the unfiltered subspace and the local variability of the stable BLVs therein.

The error in the $i$-th BLV in Eq. (28) is given by the free evolution of perturbations, formerly studied by Grudzien et al. (2018): when the filtered subspace dimension is of dimension $r \geq n_0$, we can recursively, and stably, compute the unfiltered uncertainty via

$$\widehat{\mathbf{B}}_{k+1}^{\mathrm{uu}} = \widehat{\mathbf{Q}}_{k+1}^{\mathrm{uu}} + \mathbf{U}_{k+1}^{\mathrm{uu}} \widehat{\mathbf{B}}_k^{\mathrm{uu}} \left(\mathbf{U}_{k+1}^{\mathrm{uu}}\right)^{\mathrm{T}}. \tag{29}$$

When $r < n_0$, we see explicitly that the filter will diverge as a consequence of leaving an unstable direction unfiltered.

## 3.2 Evolution of filtered error

We now consider the evolution of the projection of the forecast error into the filtered space, with respect to the reduced rank gain. From Eq. (24) we derive

$$
\begin{aligned}
\widehat{\epsilon}_{k+1}^{\mathrm{f}} = &\left(\mathbf{E}_{k+1}^{\mathrm{f}}\right)^{\mathrm{T}} \mathbf{E}_{k+1} \mathbf{U}_{k+1} \mathbf{E}_k^{\mathrm{T}} \left(\mathbf{E}_k^{\mathrm{f}} \widehat{\epsilon}_k^{\mathrm{f}} + \mathbf{E}_k^{\mathrm{u}} \widehat{\epsilon}_k^{\mathrm{u}}\right) \\
&- \left(\mathbf{E}_{k+1}^{\mathrm{f}}\right)^{\mathrm{T}} \mathbf{E}_{k+1} \mathbf{U}_{k+1} \mathbf{I}_{n \times r} \widehat{\mathbf{K}}_k \mathbf{H}_k \left(\mathbf{E}_k^{\mathrm{f}} \widehat{\epsilon}_k^{\mathrm{f}} + \mathbf{E}_k^{\mathrm{u}} \widehat{\epsilon}_k^{\mathrm{u}}\right) \\
&+ \left(\mathbf{E}_{k+1}^{\mathrm{f}}\right)^{\mathrm{T}} \left(\mathbf{E}_{k+1} \mathbf{U}_{k+1} \mathbf{I}_{n \times r} \widehat{\mathbf{K}}_k \mathbf{v}_k - \mathbf{w}_{k+1}\right).
\end{aligned}
\tag{30}
$$

Similar to Eq. (25), we see that the terms

$$
\left(\mathbf{E}_{k+1}^{\mathrm{f}}\right)^{\mathrm{T}} \mathbf{E}_{k+1} \mathbf{U}_{k+1} \mathbf{E}_k^{\mathrm{T}} \mathbf{E}_k^{\mathrm{f}} = \mathbf{U}_{k+1}^{\mathrm{ff}},
\tag{31}
$$

$$
\left(\mathbf{E}_{k+1}^{\mathrm{f}}\right)^{\mathrm{T}} \mathbf{E}_{k+1} \mathbf{U}_{k+1} \mathbf{E}_k^{\mathrm{T}} \mathbf{E}_k^{\mathrm{u}} = \mathbf{U}_{k+1}^{\mathrm{fu}},
\tag{32}
$$

using the orthogonality of the BLVs. Therefore, substitution into Eq. (24) yields

$$
\widehat{\epsilon}_{k+1}^{\mathrm{f}} = \left(\mathbf{U}_{k+1}^{\mathrm{ff}} - \mathbf{U}_{k+1}^{\mathrm{ff}} \widehat{\mathbf{K}}_k \mathbf{H}_k \mathbf{E}_k^{\mathrm{f}}\right) \widehat{\epsilon}_k^{\mathrm{f}}
\tag{33a}
$$

$$
+ \mathbf{U}_{k+1}^{\mathrm{ff}} \widehat{\mathbf{K}}_k \mathbf{v}_k - \widehat{\mathbf{w}}_{k+1}^{\mathrm{f}}
\tag{33b}
$$

$$
+ \left(\mathbf{U}_{k+1}^{\mathrm{fu}} - \mathbf{U}_{k+1}^{\mathrm{ff}} \widehat{\mathbf{K}}_k \mathbf{H}_k \mathbf{E}_k^{\mathrm{u}}\right) \widehat{\epsilon}_k^{\mathrm{u}}.
\tag{33c}
$$

The terms (33a) and (33b) correspond to the standard recursion on the KF forecast error. If the filtered subspace is the entire state space $\left(\text{i.e., } \mathbf{E}_k^{\mathrm{f}} \triangleq \mathbf{E}_k\right)$ the term (33c) is identically zero, and the terms (33a) and (33b) are equivalent to a change of basis for the forecast error recursion in Eq. (16), written in the invariant dynamics for the moving frame of the BLVs.

For $r < n$, the remaining term (33c) is our primary object of interest. Term (33c) is fundamentally different from the relationship described by terms (33a) and (33b), which represents the usual stabilizing effect of the forecast-analysis cycle. Instead, term (33c) describes two different processes: (i) $\mathbf{U}_{k+1}^{\mathrm{fu}}$ represents the purely dynamical upwelling of the unfiltered error into the filtered variables; (ii) $\mathbf{U}_{k+1}^{\mathrm{ff}} \widehat{\mathbf{K}}_k \mathbf{H}_k \mathbf{E}_k^{\mathrm{u}}$ is the correction in the filtered subspace, due to the sensitivity of these variables to observations in the unfiltered subspace, forward propagated to time $t_{k+1}$. When $\mathbf{K}_k$ yields the restricted Bayesian update, i.e., when $\mathbf{K}_k$ is defined as in Eq. (22) and $\mathbf{H}_k \mathbf{E}_k^{\mathrm{u}} \equiv \mathbf{0}_{d \times (n-r)}$, term (33c) represents dynamical upwelling alone. Generically $\mathbf{U}_{k+1}^{\mathrm{fu}} - \mathbf{U}_{k+1}^{\mathrm{ff}} \widehat{\mathbf{K}}_k \mathbf{H}_k \mathbf{E}_k^{\mathrm{u}} \neq \mathbf{0}_{r \times (n-r)}$ and $\widehat{\epsilon}_k^{\mathrm{u}}$ is Gaussian distributed with covariance given by Eq. (28), and thus is almost surely non-zero. This demonstrates that the forecast error in the filtered subspace depends on the unfiltered error via the forward evolution, whereas the unfiltered error does not depend on the error in the filtered space.

This implies that the direct application of EKF-AUS from perfect dynamics (Trevisan and Palatella, 2011b) to a linear system with model error systematically underestimates the uncertainty in the span of the leading $r$ BLVs. Specifically, EKF-AUS neglects the injection of the errors from the trailing vectors, $\widehat{\epsilon}_k^{\mathrm{u}}$, into the forecast of the leading vectors $\widehat{\epsilon}_{k+1}^{\mathrm{f}}$, represented in Eq. (33c). Even when the uncertainty in the stable BLVs is bounded uniformly (Grudzien et al., 2018), error in the trailing BLVs moves up the Lyapunov filtration, and may cause filter divergence. In perfect, linear models, where uncertainty in the

stable BLVs vanishes exponentially, the injection of error from the stable BLVs into the unstable subspace results in temporary mis-estimation though does not pose an issue to the asymptotic stability (Bocquet et al., 2017). However, with model error, the term (33c) demonstrates that reduced rank Kalman filters must be augmented to correct a persistent underestimation.

It is important to note that the error in the unfiltered subspace moves upward through the backward Lyapunov filtration precisely because the unfiltered subspace is defined by the span of the trailing BLVs, governed by the invariant upper triangular dynamics. The span of the trailing BLVs is not equal to the direct sum of the trailing Oseledec spaces, which are themselves covariant with the dynamics. This choice for the unfiltered subspace comes naturally, however, as the filtered subspace (the image space of $\mathbf{K}_k$) is given by the span of the leading BLVs, and is equivalent to the span of the leading covariant Lyapunov vectors (Kuptsov and Parlitz, 2012, see their Eq. (43)).

In principle, data assimilation could be designed to prevent dynamical upwelling of unfiltered error by defining the unfiltered space to be the direct sum of the trailing, stable Oseledec spaces. In this case, the unfiltered error would be covariant with the dynamics and leave the filtered error unaffected. To achieve this, the filtered space would need to be defined by the orthogonal complement to trailing Oseledec spaces, i.e., the span of the leading forward Lyapunov vectors (Kuptsov and Parlitz, 2012, see their Eq. (43)). However, the span of the leading forward Lyapunov vectors has been shown numerically not to contain the largest mass of the uncertainty (Ng et al., 2011). Similarly, uniformly completely observing the leading $d \geq n_0$ forward Lyapunov vectors has been shown numerically to put a weaker constraint on the growth of the uncertainty than uniformly completely observing the leading $d \geq n_0$ BLVs (Grudzien et al., 2018). Furthermore, the forward Lyapunov vectors are defined by the recursive QL factorization (Kuptsov and Parlitz, 2012), and the lower triangular dynamics for the forecast error would transmit filtered uncertainty to the unfiltered subspace, creating a dynamic downwelling which cannot be accounted for in the ensemble subspace. These results suggest that it is preferable that the unfiltered space is equal to the span of the trailing BLVs, or equivalently, that the filtered space is defined equal to the span of the leading covariant/ backward Lyapunov vectors.

With the recursive form of the filtered error in Eq. (33), we directly compute the covariance of the filtered error, and the cross covariance of the filtered and unfiltered error, in the basis of BLVs. We define the operators

$$\boldsymbol{\Phi}_{k+1} \triangleq \mathbf{U}_{k+1}^{\mathrm{fu}} - \mathbf{U}_{k+1}^{\mathrm{ff}} \widehat{\mathbf{K}}_k \mathbf{H}_k \mathbf{E}_k^{\mathrm{u}}, \tag{34}$$

$$\boldsymbol{\Sigma}_k \triangleq \left[\mathbf{I}_r - \widehat{\mathbf{K}}_k \mathbf{H}_k \mathbf{E}_k^{\mathrm{f}}\right] \widehat{\mathbf{B}}_k^{\mathrm{ff}} \left[\mathbf{I}_r - \widehat{\mathbf{K}}_k \mathbf{H}_k \mathbf{E}_k^{\mathrm{f}}\right]^{\mathrm{T}} + \widehat{\mathbf{K}}_k \mathbf{R}_k \widehat{\mathbf{K}}_k^{\mathrm{T}}, \tag{35}$$

where $\boldsymbol{\Phi}_k$ is the operator which describes the propagation of unfiltered error into the filtered space and the operator $\boldsymbol{\Sigma}_k$ corresponds to the analysis error covariance for the standard KF, written in the basis of BLVs.

We first consider the recursion for the cross covariance. In particular, by combining Eq. (33) and Eq. (27), we obtain

$$\widehat{\mathbf{B}}_{k+1}^{\mathrm{fu}} = \boldsymbol{\Phi}_{k+1} \widehat{\mathbf{B}}_k^{\mathrm{uu}} \left(\mathbf{U}_{k+1}^{\mathrm{uu}}\right)^{\mathrm{T}} + \widehat{\mathbf{Q}}_{k+1}^{\mathrm{fu}} + \mathbf{U}_{k+1}^{\mathrm{ff}} \left(\mathbf{I}_r - \widehat{\mathbf{K}}_k \mathbf{H}_k \mathbf{E}_k^{\mathrm{f}}\right) \widehat{\mathbf{B}}_k^{\mathrm{fu}} \left(\mathbf{U}_{k+1}^{\mathrm{uu}}\right)^{\mathrm{T}}. \tag{36}$$

We now consider the covariance of the forecast error in the filtered variables. Using the identity in Eq. (35) we derive the recursion for the filtered error covariance $\widehat{\mathbf{B}}_{k+1}^{\mathrm{ff}}$ as

$$\mathbf{B}_{k+1}^{\mathrm{ff}} = \mathbf{U}_{k+1}^{\mathrm{ff}} \mathbf{\Sigma}_k \left(\mathbf{U}_{k+1}^{\mathrm{ff}}\right)^{\mathrm{T}} + \widehat{\mathbf{Q}}_{k+1}^{\mathrm{ff}} \tag{37a}$$

$$+ \mathbf{\Phi}_{k+1} \widehat{\mathbf{B}}_k^{\mathrm{uu}} \mathbf{\Phi}_{k+1}^{\mathrm{T}} \tag{37b}$$

$$+ \mathbf{U}_{k+1}^{\mathrm{ff}} \left[\mathbf{I}_r - \widehat{\mathbf{K}}_k \mathbf{H}_k \mathbf{E}_k^{\mathrm{f}}\right] \widehat{\mathbf{B}}_k^{\mathrm{fu}} \mathbf{\Phi}_{k+1}^{\mathrm{T}} \tag{37c}$$

$$+ \mathbf{\Phi}_{k+1} \widehat{\mathbf{B}}_k^{\mathrm{uf}} \left[\mathbf{I}_r - \widehat{\mathbf{K}}_k \mathbf{H}_k \mathbf{E}_k^{\mathrm{f}}\right]^{\mathrm{T}} \left(\mathbf{U}_{k+1}^{\mathrm{ff}}\right)^{\mathrm{T}}. \tag{37d}$$

When the filtered space is the whole space, i.e., $\mathbf{E}_k^{\mathrm{f}} = \mathbf{E}_k$, the term (37a) entirely describes the evolution of the forecast error in the basis of BLVs — this is indeed just the forward propagation of the analysis error covariance for the KF. The term (37b) represents the contribution of uncertainty from the unfiltered subspace, propagated via the $\mathbf{\Phi}_k$ operator, while terms (37c) and (37d) describe the forward evolution of the cross covariances of the uncertainty, into the filtered space.

### 3.3  Assimilation in the unstable subspace exact (AUSE)

Having derived the exact error covariance associated to the reduced rank Kalman estimator, characteristic of the ensemble-based Kalman gain in geophysical models, we will summarize the result.

**Definition 4.** *For all $k$, let the matrix $\mathbf{B}_k$ be decomposed as in Eq.* (20). *Then, define the recursive relationship*

$$\widehat{\mathbf{B}}_k^{\mathrm{uu}} = \widehat{\mathbf{Q}}_k^{\mathrm{uu}} + \mathbf{U}_k^{\mathrm{uu}} \widehat{\mathbf{B}}_{k-1}^{\mathrm{uu}} \left(\mathbf{U}_k^{\mathrm{uu}}\right)^{\mathrm{T}}, \tag{38a}$$

$$\mathbf{\Phi}_{k+1} = \mathbf{U}_{k+1}^{\mathrm{fu}} - \mathbf{U}_{k+1}^{\mathrm{ff}} \widehat{\mathbf{K}}_k \mathbf{H}_k \mathbf{E}_k^{\mathrm{u}}, \tag{38b}$$

$$\widehat{\mathbf{B}}_{k+1}^{\mathrm{fu}} = \mathbf{\Phi}_{k+1} \widehat{\mathbf{B}}_k^{\mathrm{uu}} \left(\mathbf{U}_{k+1}^{\mathrm{uu}}\right)^{\mathrm{T}} + \widehat{\mathbf{Q}}_{k+1}^{\mathrm{fu}} + \mathbf{U}_{k+1}^{\mathrm{ff}} \left(\mathbf{I}_r - \widehat{\mathbf{K}}_k \mathbf{H}_k \mathbf{E}_k^{\mathrm{f}}\right) \widehat{\mathbf{B}}_k^{\mathrm{fu}} \left(\mathbf{U}_{k+1}^{\mathrm{uu}}\right)^{\mathrm{T}}, \tag{38c}$$

$$\mathbf{\Sigma}_k = \left[\mathbf{I}_r - \widehat{\mathbf{K}}_k \mathbf{H}_k \mathbf{E}_k^{\mathrm{f}}\right] \widehat{\mathbf{B}}_k^{\mathrm{ff}} \left[\mathbf{I}_r - \widehat{\mathbf{K}}_k \mathbf{H}_k \mathbf{E}_k^{\mathrm{f}}\right]^{\mathrm{T}} + \widehat{\mathbf{K}}_k \mathbf{R}_k \widehat{\mathbf{K}}_k^{\mathrm{T}}, \tag{38d}$$

$$\widehat{\mathbf{B}}_{k+1}^{\mathrm{ff}} = \mathbf{U}_{k+1}^{\mathrm{ff}} \mathbf{\Sigma}_k \left(\mathbf{U}_{k+1}^{\mathrm{ff}}\right)^{\mathrm{T}} + \widehat{\mathbf{Q}}_{k+1}^{\mathrm{ff}} + \mathbf{\Phi}_{k+1} \widehat{\mathbf{B}}_k^{\mathrm{uu}} \mathbf{\Phi}_{k+1}^{\mathrm{T}}$$
$$+ \mathbf{U}_{k+1}^{\mathrm{ff}} \left[\mathbf{I}_r - \widehat{\mathbf{K}}_k \mathbf{H}_k \mathbf{E}_k^{\mathrm{f}}\right] \widehat{\mathbf{B}}_k^{\mathrm{fu}} \mathbf{\Phi}_k^{\mathrm{T}} + \mathbf{\Phi}_k \widehat{\mathbf{B}}_k^{\mathrm{uf}} \left[\mathbf{I}_r - \widehat{\mathbf{K}}_k \mathbf{H}_k \mathbf{E}_k^{\mathrm{f}}\right]^{\mathrm{T}} \left(\mathbf{U}_{k+1}^{\mathrm{ff}}\right)^{\mathrm{T}}, \tag{38e}$$

*to be the Kalman Filter, Assimilation in the Unstable Subspace Exact (**KF-AUSE**) Riccati equation, for a filtered subspace of dimension $1 \leq r < n$.*

**Proposition 1.** *Assume that a Gaussian prior distribution is given for $\mathbf{x}_0$, the state of the system defined by Eq. (6). Assume that the initial uncertainty, $\epsilon_0$, is of mean zero and covariance $\mathbf{B}_0$, and suppose observations of the state are given as in Eq. (6). Let $\mathbf{K}_k$ be defined as the Kalman estimator restricted to the span of $\mathbf{E}_k^{\mathrm{f}}$ (rank $1 \leq r < n$) as in Eq. (22). Then, the forecast error defined by Eq. (16) is Gaussian, mean zero, with covariance matrix defined recursively by the KF-AUSE Riccati equation, Eq. (38).*

*Proof.* Proving the covariance is given by Eq. (38) is the content of sections 3.1 and 3.2. That the error is mean zero and Gaussian is easily proven by induction. □

It should be noted that the KF-AUSE Riccati equation is also valid for the exact forecast error covariance of a reduced rank Kalman filter in perfect models, where $\mathbf{Q}_k \triangleq \mathbf{0}_n$ for all $k$. Let $r = n_0$, $\mathbf{Q}_k \triangleq \mathbf{0}_n$ and $\mathbf{P}_k \triangleq \mathbf{E}_k^{\mathrm{f}} \boldsymbol{\Gamma}_k \left( \mathbf{E}_k^{\mathrm{f}} \right)^{\mathrm{T}}$ be defined as the estimated forecast error covariance for EKF-AUS (Trevisan and Palatella, 2011b), then the recursion is defined by

$$\boldsymbol{\Gamma}_{k+1} \triangleq \mathbf{U}_{k+1}^{\mathrm{ff}} \left[ \mathbf{I}_r - \widehat{\mathbf{K}}_k \mathbf{H}_k \mathbf{E}_k^{\mathrm{f}} \right] \boldsymbol{\Gamma}_k \left[ \mathbf{I}_r - \widehat{\mathbf{K}}_k \mathbf{H}_k \mathbf{E}_k^{\mathrm{f}} \right]^{\mathrm{T}} \left( \mathbf{U}_{k+1}^{\mathrm{ff}} \right)^{\mathrm{T}} + \mathbf{U}_{k+1}^{\mathrm{ff}} \widehat{\mathbf{K}}_k \mathbf{R}_k \widehat{\mathbf{K}}_k^{\mathrm{T}} \left( \mathbf{U}_{k+1}^{\mathrm{ff}} \right)^{\mathrm{T}}, \tag{39}$$

analogous to term (37a). Comparing Eq. (38) and Eq. (39), we see that even in perfect models the estimated error covariance of EKF-AUS in the filtered subspace and the exact error covariance do not agree, i.e., $\boldsymbol{\Gamma}_{k+1} \neq \widehat{\mathbf{B}}_{k+1}^{\mathrm{ff}}$. This is because the estimated AUS error in Eq. (39) neglects the upwelling of the initial error in the unfiltered subspace, described by terms (37b), (37c) and (37d). However, the unfiltered error decays exponentially and the mis-estimation in the filtered space does not threaten filter stability: the AUS estimated error covariance converges to the exact error in its asymptotic limit, though possibly arithmetically (Bocquet et al., 2017).

### 3.4 Discussion: dynamical upwelling and covariance inflation

We emphasize that the KF-AUSE Riccati equation (38) is not intended to provide a computational advantage — its computation requires knowledge of error in the unfiltered subspace and, in nonlinear models, a full rank representation of the tangent linear dynamics. Nonetheless, this recursion is demonstrative of an important concept: for a reduced rank Kalman estimator that applies its analysis update in the span of the leading BLVs, the exact error in the same span always depends on the unfiltered error in the trailing vectors. This dependence is described explicitly by the terms (37b) - (37d) for the recursion on the filtered error covariance in KF-AUSE, representing the missing terms in the Kalman filter recursion necessary to describe the exact uncertainty in the ensemble span.

The upwelling of uncertainty from the unfiltered subspace to the ensemble span thus highlights a dynamical mechanism, and provides a theoretical motivation for why covariance inflation in the EnKF has been successful in preventing filter divergence. In certain scenarios, due to the neglected upwelling terms in the standard Kalman filter error recursion, covariance inflation may emulate the process of upwelling in the ensemble span, replicating the increased uncertainty in the ensemble span due to the injection of terms (37b) - (37d), or otherwise ameliorate the effect of neglecting these terms.

Generally, the reasons for using covariance inflation in the EnKF are wide, including treatment of model error, sampling error, intrinsic bias, and non-Gaussianity of error distributions (Raanes et al., 2018, see section 2.2 for a survey). However, Eq. (38) demonstrates that even when excluding nonlinearity, non-Gaussianity, and intrinsic deficiencies of the EnKF, the exact correction to the error in the ensemble span requires the covariance of the unfiltered error as well as the cross covariance of the error in the filtered and unfiltered subspaces, as in Eq. (38). In practice, one must find a suitable approximation of the upwelling phenomenon to prevent the systematic underestimation of the forecast error, and/or, extend the rank of the ensemble-based correction to control the transient growth of errors in the stable modes.

Reduced rank Kalman filters have previously corrected for the upwelling of model errors with both multiplicative and additive covariance inflation methods. Although it was not explicitly formulated as such, the SEEK filter of Pham et al. (1998) can been seen to compensate for model errors originating in the unfiltered, stable subspace: while components of the model

error covariance which are orthogonal to the filtered subspace are left neglected, there is an implicit treatment by utilizing its forgetting factor to inflate the variance of the estimated error in the filtered subspace (Nerger et al., 2005). The contribution of the unfiltered error to the estimated error was also studied in ensemble methods by Raanes et al. (2015), in which the authors explored sampling methodology to compensate for the unresolved model errors, residing outside of the ensemble span. Our work adds to this discussion, now highlighting the explicit mechanism which these covariance inflation techniques have compensated for.

The dynamical upwelling of model error differs from the misrepresentation of the covariance due to truncation error or sampling error induced by nonlinear dynamics in perfect models, treated in the modified EKF-AUS-NL (Palatella and Trevisan, 2015) and in the finite size ensemble Kalman filter, **(EnKF-N)** (Bocquet, 2011; Bocquet et al., 2015). We have shown that the upwelling of the unfiltered error through the Lyapunov filtration acts as a linear effect and is acute in the presence of additive model errors which are excited by transient instabilities. While the effect of the dynamical upwelling could be neglected in perfect models (Bocquet et al., 2017), the work of Grudzien et al. (2018) has demonstrated that transient instability in the span of the stable BLVs can drive the unfiltered error to become impractically large — furthermore, this error is transmitted into the filtered subspace, driving filter divergence if it is left uncorrected. However, the significance of perfect models is not lost: if the dimension of the filtered space is sufficiently large such that dynamical stability rapidly dissipates unfiltered errors, the effect of the upwelling may become negligible.

Without otherwise augmenting the ensemble-based Kalman gain, the upwelling of uncertainty into the filtered space can, in certain scenarios, be emulated with multiplicative inflation. In the following section, we numerically explore the interaction of the filtered subspace rank, the stability in the unfiltered directions, and multiplicative covariance inflation in relation to the effect of dynamical upwelling in reduced rank Kalman filters. However, while the results of section 4 empirically validate the hypothesis that multiplicative inflation can compensate for unrepresented dynamical upwelling, they also reveal how multiplicative inflation may be obviated by less ad hoc methods. Likewise, the observed structure of the reduced rank covariance suggests that, even when the upwelling of error is well parametrized, the greatest driver of forecast uncertainty may be due to the presence of unfiltered errors in the trailing BLVs — this will be the subject of the discussion in section 5.

## 4 Numerical results

### 4.1 Experimental setup

We will explore two different discrete model configurations in which we vary the effect of nonlinearity. In the continuous model configuration with stochastic differential equations, we also achieve qualitatively similar results which will not be included. It is important to remark that the analytic form for the forecast error in Eq. (38) is only a useful representation for weakly-nonlinear evolution of error, corresponding to the error evolution of the EnKF on short time scales. As the effect of nonlinearity is increased, the linear approximations utilized in our work will no longer be adequate, leading to truncation errors as discussed by, e.g., Palatella and Trevisan (2015).

In the following, we use two different formulations of the standard **Lorenz 96 equations (L96)** (Lorenz and Emanuel, 1998), commonly used in data assimilation literature see, e.g., Carrassi et al. (2018)[and references therein]. For each $m \in \{1, \cdots, n\}$, the (L96) equations read $\frac{d\mathbf{x}}{dt} \triangleq \mathbf{L}(\mathbf{x})$,

$$L^m(\mathbf{x}) = -x^{m-2}x^{m-1} + x^{m-1}x^{m+1} - x^m + F \tag{40}$$

such that the components of the vector $\mathbf{x}$ are given by the variables $x^m$ with periodic boundary conditions, $x^0 = x^n$, $x^{-1} = x^{n-1}$ and $x^{n+1} = x^1$. The term $F$ in L96 is the forcing parameter. The tangent linear model (Kalnay, 2003) is governed by the equations of the Jacobian matrix, $\nabla\mathbf{L}(\mathbf{x})$,

$$\nabla L^m(\mathbf{x}) = \left(0, \cdots, -x^{m-1}, x^{m+1} - x^{m-2}, -1, x^{m-1}, 0, \cdots, 0\right). \tag{41}$$

### 4.1.1 Discrete linear experiments

In linear experiments, we construct a discrete, linear model from the L96 system. Fixing the system dimension $n \triangleq 10$, the linear propagator in our model $\mathbf{M}_k$ is generated by computing the discrete, tangent linear model from the resolvent of the Jacobian equation, Eq. (41). In generating the discrete, tangent linear model, the discretization time between observations is fixed at $\delta_k \triangleq \delta = 0.1$ for all $k$. We numerically integrate the Jacobian equation with a fourth order Runge-Kutta scheme with a fixed time step of $h \triangleq 0.01$. For the forcing value of $F = 8$, with 10 dimensions, there are three unstable, one neutral, and six stable Lyapunov exponents, i.e, $n_0 = 4$. The observation error covariance $\mathbf{R}_k$, model error covariance $\mathbf{Q}_k$ and observation operator $\mathbf{H}_k$ are all fixed as the identity $\mathbf{I}_{10}$ in this setup for simplicity.

### 4.1.2 Discrete nonlinear experiments

In our experiments with the discrete extended Kalman filter for nonlinear systems, we use Eq. (40) directly for our model state evolution, and fix the state dimension to $n \triangleq 40$. For the 40 dimensional L96, with standard forcing $F = 8$, the unstable neutral subspace is of dimension $n_0 = 14$, with one neutral Lyapunov exponent. The nonlinear trajectory is integrated with a fourth order Runge-Kutta scheme, with a fixed step size of $h \triangleq 0.05$, and an interval between observation times of $\delta_k \triangleq \delta = 0.1$. At each observation time, before observations are given, the true trajectory is perturbed (in model space) by additive Gaussian noise with a prescribed covariance $\mathbf{Q}$, fixed in time. In general, the random model noise can be injected at different intervals than the interval between observations, affecting the nonlinearity of the error evolution. However, we fix these intervals to be equal for simplicity.

Let us define the nonlinear map $\mathbf{\Psi}(t_0, t_1) : \mathbb{R}^n \to \mathbb{R}^n$ to be the flow map, generated from Eq. (40), that takes the model state from time $t_0$ to $t_1$. Then, noting that $\mathbf{\Psi}(t, t+\delta) = \mathbf{\Psi}(s, s+\delta)$ for all $t$ and $s$, we will define $\mathbf{\Psi}_\delta \triangleq \mathbf{\Psi}(0, \delta)$. In our experiments, the "truth" is thus evolved via the equation,

$$\mathbf{x}_{k+1} = \mathbf{\Psi}_\delta(\mathbf{x}_k) + \mathbf{w}_{k+1}, \tag{42}$$

$\mathbf{w}_{k+1} \sim N(\mathbf{0}, \mathbf{Q})$, while the mean trajectory of the "model" state is given by the deterministic evolution, $\mathbf{x}^{\mathfrak{b}}_{k+1} = \mathbf{\Psi}_\delta(\mathbf{x}^{\mathfrak{b}}_k)$. In our experimental design, the extended Kalman filter estimates the state of the nonlinear "true" state, perturbed by the noise $\mathbf{w}_k$, Eq. (42), and $\mathbf{M}_{k+1}$ (the linear propagator for the covariance forward evolution) is defined by the map $\mathbf{M}_{k+1} \triangleq \nabla \mathbf{\Psi}_\delta \big|_{\mathbf{x}^{\mathfrak{b}}_k}$.

The matrix $\mathbf{Q}$ is defined by the circulant matrix with $c_0 = 0.5, c_1 = 0.25, c_2 = 0.125, c_{39} = 0.25, c_{38} = 0.125$ and all other entries zero,

$$
\mathbf{Q} \triangleq \begin{pmatrix}
c_0 & c_{39} & \cdots & c_2 & c_1 \\
c_1 & c_0 & c_{39} & & c_2 \\
\vdots & c_1 & c_0 & \ddots & \vdots \\
c_{38} & & \ddots & \ddots & c_{39} \\
c_{39} & c_{38} & \cdots & c_1 & c_0
\end{pmatrix} . \tag{43}
$$

The choice of the circulant matrix reflects the stationary statistics and periodic nature of the L96 model, and the fact that we wish to highlight the effect of analytically resolving complex model error. The observation error covariance matrix is fixed as $0.25 * \mathbf{I}_{40}$. The observation operator is fixed in time as $\mathbf{H}_k \triangleq \mathbf{I}_{40}$.

This experimental configuration is mathematically consistent with the extended Kalman filter for a discrete nonlinear map with model error, and is a standard formulation for model error twin experiments, utilized by e.g, Mitchell and Carrassi (2015); Sakov et al. (2018), with the configuration using the circulant covariance matrix, $\mathbf{Q}$, drawn specifically from Raanes et al. (2015). The interval between observations $\delta$ controls the nonlinearity of evolution of the forecast errors in the combined forecast/analysis data assimilation cycle. Our chosen configuration for the observation interval, and the interval of random forcing in the model, can be considered weakly-nonlinear.

### 4.2 Linear Kalman filter

In a linear setting, we compute the exact forecast error covariance of KF-AUSE via the recursive Riccati equation, Eq. (38), and compare it with that of the KF, for which the filtered space is the entire model space. This illustrates the performance of a rank deficient filter where the forecast error is treated analytically, without mis-estimation of the error covariances. We compute the average eigenvalues of the forecast covariance matrix for each the KF and KF-AUSE over 100,000 parallel forecast cycles and examine the stratification of the uncertainty in a basis of BLVs, i.e., how strongly the covariance projects into each direction. Specifically, for both the KF and KF-AUSE we compute the average projection coefficient of the forecast error covariance into the $i$-th BLV at each forecast time, $\left(\mathbf{E}^i_k\right)^{\mathrm{T}} \mathbf{B}_k \mathbf{E}^i_k$, and average this coefficient over $k$.

In Fig. 1, the averaged eigenvalues of the KF and KF-AUSE forecast error covariance are plotted, with triangle markers, differentiated by color. In each subplot, the KF remains the same but we vary the dimension of the filtered subspace, $r$, for KF-AUSE. In the top left panel of Fig. 1 the number of corrected modes is equal to $n_0$, corresponding to correcting the error in the unstable-neutral subspace. Here, the leading eigenvalue of the forecast uncertainty of KF-AUSE is orders of magnitude above the forecast uncertainty in the KF. This should be contrasted with perfect models where, asymptotically, there can only be four non-zero eigenvalues, and under generic conditions, the KF and EKF-AUS will coincide (Bocquet et al., 2017). In

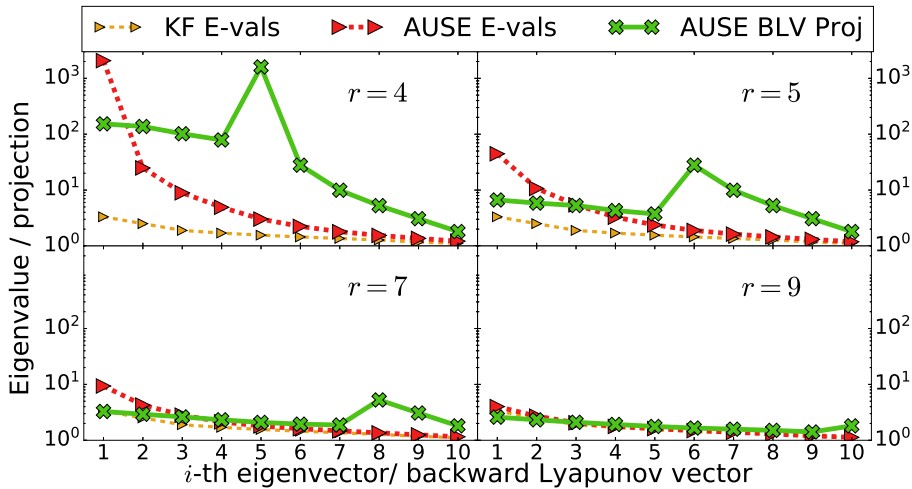

**Figure 1.** Eigenvalues of the KF and KF-AUSE forecast error covariance plotted with triangles. Projection coefficients of the KF-AUSE forecast error covariance plotted with X's. Dimension of the KF-AUSE filtered subspace is $r$. Note the log scale of the $y$-axis.

accordance with the results of Grudzien et al. (2018), correcting error in the first stable mode ($r = 5$) brings a substantial reduction in forecast uncertainty (see top right Fig. 1). We see the forecast uncertainty likewise diminishes as each additional mode is corrected, as the KF-AUSE covariance converges to that of the KF.

It is of special interest how the projection coefficients of the forecast error covariance relates to the dimension of the filtered subspace, $r$. In the KF, the projection coefficients are closely aligned with the eigenvalue profile, descending in the order of the Lyapunov exponents, and this line is not pictured due to the redundancy. However, in the forecast error covariance of KF-AUSE, the leading uncorrected stable mode is the dominant direction for the uncertainty among the BLVs, systematically across $n_0 \le r < n$, with projection coefficient on the order of the leading eigenvalue. This distinguishes the setting of additive model error from perfect models where the projection coefficients of the forecast error covariance in the stable BLVs will be

zero asymptotically (Gurumoorthy et al., 2017).

### 4.3   Discrete extended Kalman filter

In our experiments with the discrete extended Kalman filter, we compute the analysis root mean square error **(RMSE)** of each the: (i) full rank extended Kalman filter **(EKF)**, (ii) EKF-AUS and (iii) EKF-AUSE, for which Eq. (38) is used to compute the estimated covariance and rank $r$ gain. We will study the effect of analytically resolving the unfiltered error as compared

with the straightforward implementation of EKF-AUS, which will make no correction to account for the unfiltered error in the trailing BLVs, or its upwelling into the leading BLVs.

     Recall that EKF-AUS has historically only been studied without additive model errors — we implement EKF-AUS in the presence of model error by computing a rank $r$ estimated error covariance, which includes the projection of the model error covariance, $\mathbf{Q}_k$ into the span of the leading BLVs in the forecast Riccati equation, i.e. $\left(\mathbf{E}_k^{\mathrm{f}}\right)^{\mathrm{T}} \mathbf{Q}_k \mathbf{E}_k^{\mathrm{f}} = \widehat{\mathbf{Q}}_k^{\mathrm{ff}}$. This corresponds to

utilizing only the first line of the recursion for $\widehat{\mathbf{B}}_k^{\text{ff}}$, Eq. (37a), to compute the estimated forecast error covariance of EKF-AUS. The implementation of EKF-AUSE thus differs by utilizing a full rank ensemble of anomalies to compute the complete Riccati equation, Eq. (38).

We study the performance of EKF-AUS/E when the dimension of the filtered subspace is greater than, or equal to, the dimension of the unstable-neutral subspace; the case $r < n_0$ will trivially lead to divergence (Bocquet et al., 2017). In Fig. 2, we plot the analysis RMSE of EKF-AUS and EKF-AUSE with triangles and X's respectively, while we vary over the dimension of the filtered subspace, with the RMSE computed over 100,000 analysis cycles.

To benchmark the performance of EKF-AUS/E, we plot the observation error standard deviation and the analysis RMSE of the standard, full rank EKF in horizontal lines — the algorithms for EKF-AUS/E are tantamount to a change of basis for the EKF when the filtered subspace is equal to the full space, and thus this is the logical point of comparison. We are interested in finding the necessary dimension of the filtered subspace such that EKF-AUS/E has an RMSE which: (i) performs better than the observation error standard deviation and (ii) performs comparably to filtering the entire space. When the RMSE of EKF-AUS/E falls below the observation error standard deviation, the filter has a forecast performance superior to initializing observations directly in the model; when it performs closely to the EKF, the filter can be considered close to optimal performance, while utilizing a sub-optimal correction based on only $r < n$ directions.

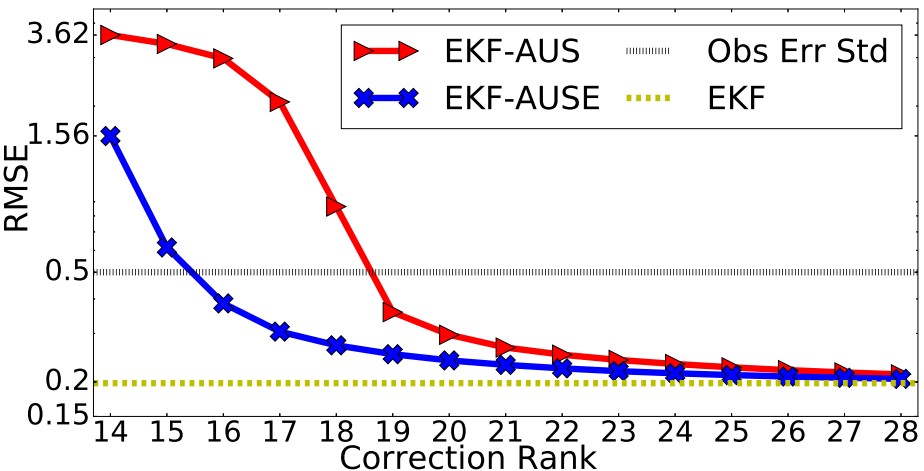

**Figure 2.** Analysis RMSE of EKF-AUS plotted with triangles and EKF-AUSE plotted with X's, varying over the rank of the sub-optimal gain. Horizontal lines are the observation error standard deviation and EKF analysis RMSE. Note the log scale of the $y$-axis.

In Fig. 2, when the dimension of the filtered subspace for both AUS/E reaches 28 the difference between both EKF-AUS/E and the full-rank EKF becomes negligible. The RMSE of the: (i) EKF is approximately 0.198; (ii) EKF-AUS, $r = 28$, is approximately 0.213; (iii) EKF-AUSE, $r = 28$, is approximately 0.205. The fact that EKF-AUS obtains near optimal performance, representing the uncertainty in the leading $r = 28$ BLVs while neglecting the remaining, corroborates the claim of Grudzien et al. (2018): in the presence of model noise, the filter correction should also incorporate weakly stable directions that can

be instantaneously unstable. It is of particular interest, however, that the convergence of EKF-AUSE to the skill of the full rank EKF is substantially faster: EKF-AUSE obtains adequate filter performance (RMSE lower than observation error standard deviation) by correcting the error in only 16 BLVs while EKF-AUS requires a correction of rank 19. For other scalings of the matrix $\mathbf{Q}$, multiplying $\mathbf{Q}$ by 0.1, 0.2, 1.5, 2, changing the observation dimension, e.g. $d = 20$ or $d = 30$, and by varying

the time between observations, e.g. $\delta_k = 0.01$ or 0.5, we obtain qualitatively similar results, that are not pictured here. The profiles of the curves in Fig. 2 are similar across these experimental configurations: the RMSE of EKF-AUSE is improved over EKF-AUS by analytically resolving the effect of the analytical, and the RMSE approaches an adequate/optimal level with a smaller dimension for the filtered space. We emphasize again that EKF-AUSE does not represent a computational advantage as a full rank set of perturbations is used to describe the analytic form for the upwelling of the error.

We look at the behavior of the local Lyapunov exponents for the L96 model to explain the convergence of EKF-AUS to the full rank EKF. In Fig. 3 we show the box plot statistics of the local Lyapunov exponents for exponents 14 through 28 of the L96 model. Exponent $\lambda_{14} = 0$, and the remaining pictured exponents correspond to the leading, stable BLVs. We emphasize that the local Lyapunov exponents of $\lambda_{15}$ through $\lambda_{18}$, though having negative mean, are sufficiently unstable locally such that EKF-AUS diverges when it disregards the upwelling of the error from these asymptotically stable modes.

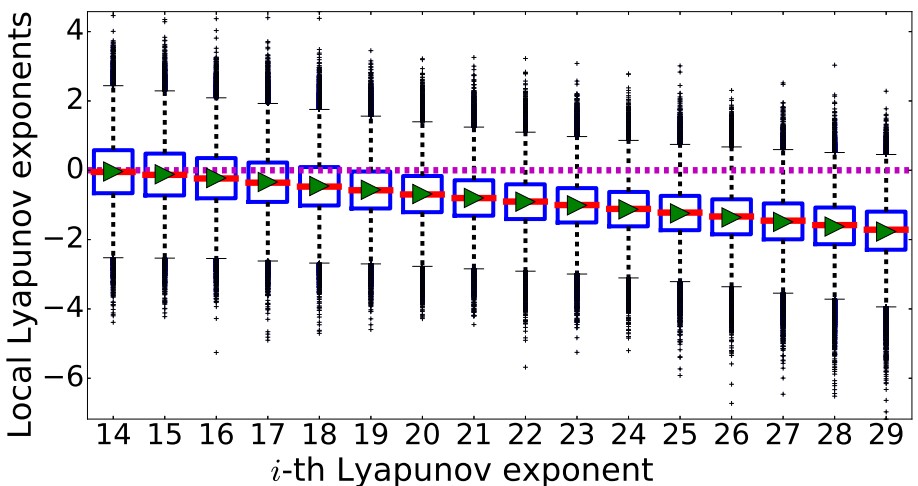

**Figure 3.** Box plot statistics of the local Lyapunov exponents, for Lyapunov exponents 14 through 24, over 100,000 realizations for the 40 dimensional L96 model. The mean ($i$-th Lyapunov exponent) is plotted as a triangle with median the horizontal line. Box contains inner two quartiles of realizations, with whiskers extending to 1.5 the inner quartile width from the third and first quartile. Outliers are realizations outside of this range, plotted individually.

When the filtered subspace for EKF-AUS is of dimension 19, such that the leading unfiltered BLV corresponds to $\lambda_{20}$, all unfiltered Lyapunov exponents have over $75\%$ of local realizations strictly stable; this corresponds to the rank when EKF-AUS has adequate performance. Likewise, the difference between EKF-AUS/E and the EKF is negligible when the leading unfiltered BLV corresponds to $\lambda_{29}$, with only $1.51\%$ of its local realizations being non-negative. These findings are consistent with the

results in Grudzien et al. (2018): in the presence of model error, unconstrained forecast error is strongly forced by the error in BLVs, which are asymptotically stable but, that experience strong and frequent local instabilities.

Finally, we are interested in how analytically computing the upwelling of error from the unfiltered subspace, as in EKF-AUSE, compares with a homogeneous, multiplicative inflation applied to the EKF-AUS algorithm. Multiplicative scalar inflation is among the most common approaches to mitigate for sampling and model error in Kalman filtering methods, and it is widely used in operational environmental forecasts utilizing the EnKF. We define $\mathbf{P}_k \triangleq \left(\mathbf{E}_k^{\mathrm{ff}}\right)^{\mathrm{T}} \left(\mathbf{\Gamma}_k + \widehat{\mathbf{Q}}_k^{\mathrm{ff}}\right) \mathbf{E}_k^{\mathrm{ff}}$ to be the estimated forecast error of EKF-AUS, where $\mathbf{\Gamma}_k$ is defined in Eq. (39). The inflated covariance $\mathbf{P}_k^{\mathrm{I}}$ is defined as $\mathbf{P}_k^{\mathrm{I}} = \left(\mathbf{E}_k^{\mathrm{ff}}\right)^{\mathrm{T}} \left(\alpha \mathbf{\Gamma}_k + \widehat{\mathbf{Q}}_k^{\mathrm{ff}}\right) \mathbf{E}_k^{\mathrm{ff}}$ for some chosen scalar $\alpha$. The inflated covariance $\mathbf{P}_k^{\mathrm{I}}$ is used to compute the reduced rank gain, as a simple way to compensate for the underestimation of the forecast error when using the recursion in Eq. (37a). Furthermore, the inflated covariance is subsequently used in the recursion for the subsequent analysis and forecast error covariances.

From the results in Fig. 2, we select the dimension of the filtered subspace to be 17, such that EKF-AUSE has RMSE below the observation error standard deviation while EKF-AUS (without inflation) has diverged. In Fig. 4, we plot the analysis RMSE of EKF-AUSE, with filtered subspace dimension 17, the observation error standard deviation and the full-rank EKF analysis RMSE as in Fig. 2 as horizontal lines. Additionally, we plot the analysis RMSE (y-axis) of EKF-AUS as a function of the inflation value (the x-axis) applied to the forecast error covariance. The inflation values, $\alpha$, are defined as the evenly spaced points in $[1, 4]$ at increments of $0.1$, denoted by triangles. The RMSE is again computed over 100,000 forecast cycles.

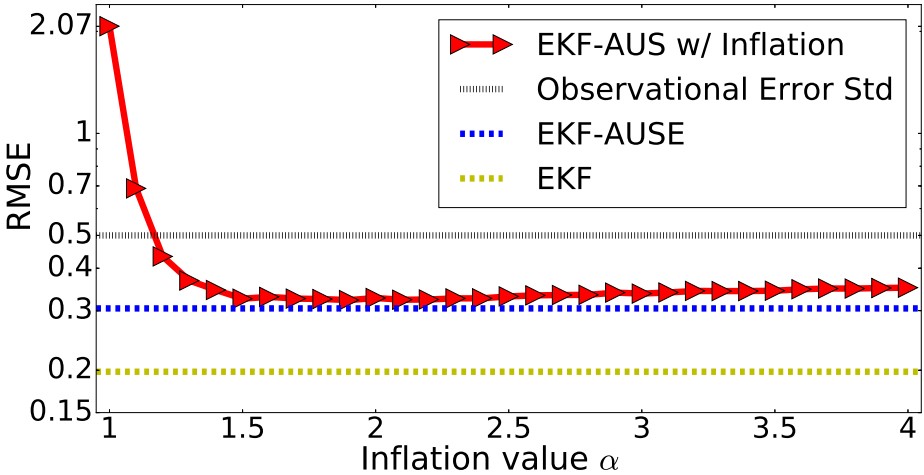

**Figure 4.** Analysis RMSE of EKF-AUS (y-axis), correction rank 17, with multiplicative inflation plotted versus the inflation value $\alpha$ (x-axis). Horizontal lines are the observation error standard deviation, EKF-AUSE and EKF analysis RMSE. Note the log scale of the $y$-axis.

Figure 4 highlights distinctly the impact of including multiplicative inflation to EKF-AUS: the performance of EKF-AUS with inflation quickly becomes comparable to the analytically resolved EKF-AUSE, which in this case, represents the lowermost bound for the RMSE of EKF-AUS with homogeneous inflation. The lowest RMSE for EKF-AUS with inflation, realized in Fig. 4, is approximately 0.322 compared to the RMSE of EKF-AUSE, approximately 0.304. Figure 4 confirms the role

of multiplicative inflation as compensating for the upwelling of unfiltered error under weakly-nonlinear error growth, and explains the underlying dynamical mechanism: multiplicative inflation brings the estimated forecast error covariance of EKF-AUS closer to the covariance given by EKF-AUSE.

## 5    Discussion: the reduced rank KF covariance and gain augmentation

Whitaker and Hamill (2012) found evidence that additive inflation could better compensate for the effects of unresolved model error, while multiplicative inflation is best suited to account for sampling error, consistent with what was noted by Bocquet (2011) and Bocquet and Sakov (2012). This hypothesis is supported by our results as follows. The combination of rank deficiency of the analysis and the presence of additive model error leads to a persistent, residual unfiltered uncertainty, and its resultant upwelling into the ensemble span of the EnKF. The dynamical upwelling forms the basis for a systematic underestimation of the uncertainty in the ensemble space, as demonstrated in Fig. 2. This can be compensated for with multiplicative inflation in the ensemble span, which emulates the additional uncertainty that is neglected in the standard, reduced rank Kalman filter recursion — this effect is exhibited in Fig. 4. Figure 5 gives a conceptual diagram of the number of samples (ensemble members) needed to prevent divergence of the EnKF in different dynamical regimes, and the effect of multiplicative inflation on this requirement.

However, multiplicative inflation (in the ensemble span) neglects the fundamental issue that the unfiltered error lying outside of the ensemble span can be the major driver of the uncertainty in a reduced rank filter with model error. Figure 1 shows that when the upwelling is analytically resolved, the largest uncertainty typically lies in the leading unfiltered BLV, even when this is an asymptotically stable mode. We provide a conceptual, two-dimensional visualization of the difference between the standard (full rank) Kalman filter forecast error covariance and the reduced rank Kalman filter forecast error covariance in Fig. 6. Note that the shape of the reduced rank Kalman filter forecast error covariance may depend strongly on the model error covariance $\mathbf{Q}$.

Generally, unless local Lyapunov exponents in the unfiltered space are strongly stable and thereby rapidly dissipate the unfiltered perturbations of model error, transient instabilities can make the unfiltered errors large enough to prevent useful state estimates (Grudzien et al., 2018). This is evidenced in Fig. 4 where neither EKF-AUSE or EKF-AUS, with multiplicative inflation, achieve an RMSE comparable with the full rank EKF. For this reason, it is highly pertinent to explore the role of augmenting the EnKF gain with a sub-optimal correction which provides some control on the transient error growth in the orthogonal complement to the ensemble span. Ideally, some constraint on the unfiltered error, even if sub-optimal, would further close the gap between the RMSE of EKF-AUSE and EKF in Fig. 4.

This issue of instability forcing unfiltered error is even more acute in practice. If an EnKF applies a correction of rank less than the number of unstable and neutral Lyapunov exponents, it has been found that the filter's estimated error can become small while the filter permanently loses track of the true trajectory (Ng et al., 2011). This behavior is easily understood in terms of the filter's failure to correct the error growth in the span of at least one of the unstable-neutral BLVs. For large geophysical models, computational limitations may prohibit the use of an ensemble of size sufficient to even span the unstable-

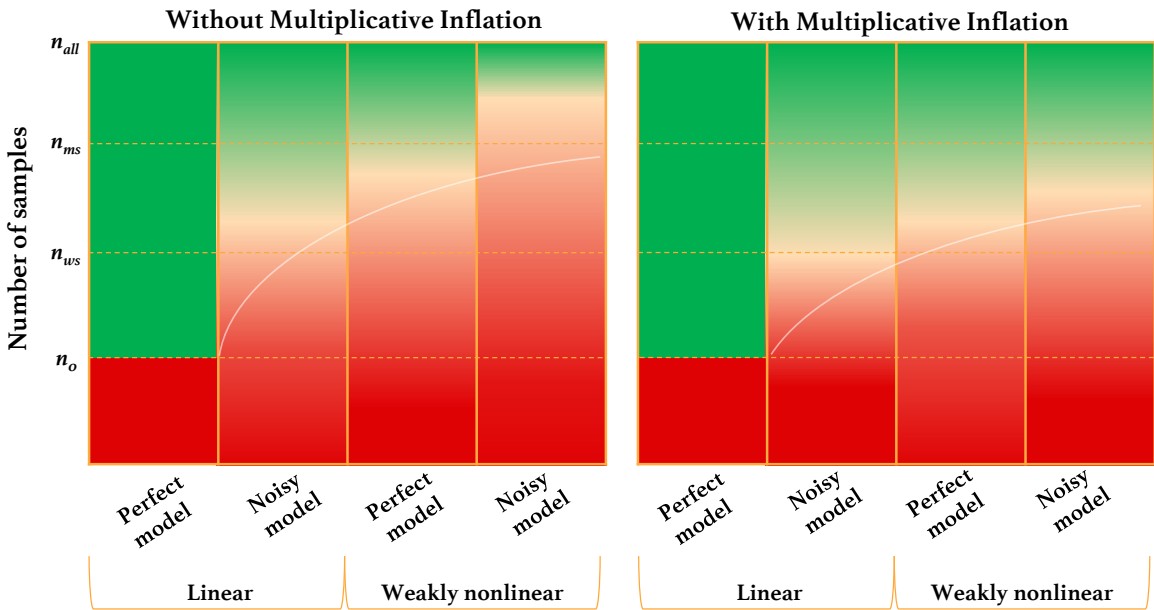

**Figure 5.** Conceptual representation of the number of samples necessary to prevent divergence of the EnKF in different filtering regimes. Dark green represents near-optimal filter performance and dark red represents filter divergence. In perfect-linear models, only $n_0$ samples are needed for an asymptotically optimal performance. Without inflation, in noisy linear and perfect, weakly-nonlinear regimes, near optimal performance can be obtained by correcting error in all modes up to the moderately stable BLVs — here $n_{ws}$ corresponds to the number of unstable/ neutra/ weakly-stable modes, while $n_{ms}$ furthermore includes moderately-stable modes. Additional samples may be necessary to control error growth with noisy, weakly-nonlinear evolution. Multiplicative inflation corrects for the upwelling from the uncorrected stable modes so that near optimal performance can be obtained when the error growth in unstable/ neutral/ weakly-stable modes are corrected.

neutral subspace, let alone the weakly stable modes which exhibit transient instabilities. In this case, the unfiltered error in the unstable-neutral modes can grow, possibly exponentially, and the filter may experience catastrophic filter divergence, due to the failure of the ensemble-based gain to correct the error in the span of all the unstable-neutral BLVs (Penny, 2017).

  Hybridization of the ensemble-based gain and additive inflation of the ensemble-based covariance are two historical methods
5 for compensating for the inability to correct for instabilities outside of the ensemble span. In hybridization, the ensemble-based Kalman estimator is augmented by a static, climatologically based estimator — using a background climatological covariance, the rank of the estimator used for the analysis update is increased, and has the effect of applying a correction to additional modes outside of the ensemble span (Hamill and Snyder, 2000). Likewise, the use of additive, random perturbations to the ensemble-based covariance has been shown to prevent filter divergence by rectifying the rank deficiency of the covariance, and
10 therefore the rank deficiency of the ensemble-based gain (Corazza et al., 2007).

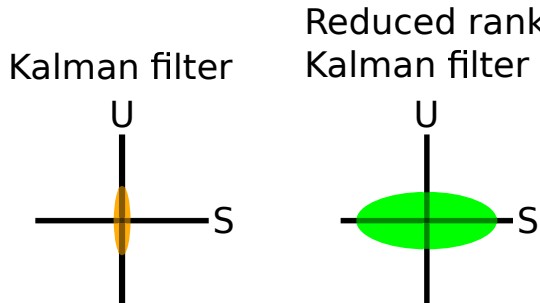

**Figure 6.** Conceptual digram of the shape of the exact forecast error covariance of the full rank Kalman filter and the exact reduced rank Kalman filter. The **U** axis represents the span of the unstable-neutral BLVs, where the forecast uncertainty projects most strongly in the standard (full rank) Kalman filter. The **S** axis represents the span of the stable BLVs, where the uncertainty is the largest (though bounded), for a reduced rank Kalman filter that neglects corrections to these modes. The comparison between the full rank and reduced rank Kalman filter covariance corresponds to the behavior exhibited in the curves in Fig. 1.

However, there is considerable difficulty in mathematically analyzing the exact recursive form for a sub-optimal augmentation of the ensemble-based covariance and ensemble-based Kalman gain. Although the dynamical upwelling of errors is a generic dynamical feature of these systems, the one-way dependence of the error in the leading BLVs on the trailing BLVs does not persist, due to the introduction of estimation errors into the trailing modes via the augmented gain. Moreover, the surrogate
covariance used to constrain error in the trailing BLVs will not generally agree with the exact error covariance in the trailing BLVs, making a closed form more difficult to derive. In this setting, it may be more appropriate to derive heuristic methods which attempt to: (i) provide some corrections in the trailing BLVs, albeit sub-optimal; (ii) describe the dynamical upwelling of the residual error from the trailing BLVs into the leading BLVs; and (iii) describe the cross covariances, between the leading and trailing BLVs, with respect to the corrections.
Multiplicative inflation may be used in this case to account for mis-estimation of forecast errors resulting from these approximations, but this mis-estimation can also be accounted for using less ad hoc approaches including parameterizing this error with hyperpriors (Bocquet et al., 2015). We argue that the hyperprior in the EnKF-N can, in principle, also be selected to take into account the dynamical upwelling exhibited by KF-AUSE. Recently, an extension of the EnKF-N to the presence of model error has utilized an adaptive multiplicative inflation term to compensate for model errors (Raanes et al., 2018), but we sug-
gest that an alternative approach including gain augmentation (Bocquet et al., 2015, suggested in section 7), and a hyperprior parametrizing the resulting error distribution, including dynamical upwelling, would be a logical extension for future research.

## 6  Conclusions

Assimilation in the Unstable Subspace (AUS) has provided a useful conceptual framework for understanding the dynamical properties of data assimilation cycling in perfect models. Both numerical and mathematical results have confirmed the un-
derlying hypothesis of Anna Trevisan: in the setting of perfect, chaotic models, the evolution of uncertainty is confined to a

space characterized by non-negative Lyapunov exponents, typically of much lower dimension than the full model state space (Palatella et al., 2013). In ensemble data assimilation, we see that the asymptotic characteristics of the anomalies exhibit these properties, which can be exploited to reduce the computational burden of the assimilation cycle (Bocquet and Carrassi, 2017). This phenomena has recently also been utilized to reduce the numerical cost of synchronization in dynamical shadowing based

data assimilation methods (de Leeuw et al., 2017). The work of Palatella and Grasso (2018) has furthermore proposed an extension of the EKF-AUS-NL algorithm to account for parametric model errors.

This paper now demonstrates that the framework of AUS can likewise be used to understand the underlying mechanisms for the evolution of uncertainty for ensemble-based filters in chaotic models with additive errors. Due to the high dimensional models, and unresolved physical processes, this circumstance is ubiquitous in high-dimensional geoscience applications where

standard EnKFs are extremely rank deficient. Utilizing the Lyapunov filtration for the backward vectors, we have shown how unfiltered error, outside of the span of the anomalies, is transmitted by the dynamics into the filtered subspace. In perfect models, or when stability in the unfiltered subspace is sufficiently strong, this effect can be neglected due to the rapid dissipation of unfiltered errors. However, Grudzien et al. (2018) demonstrate how weakly stable modes of high variance can go through periods of transient instability, exciting unfiltered error. The dynamic upwelling of unfiltered error, characterized by the term

(33c) in the forecast error recursion, and by the terms (37b) - (37d) in the filtered error covariance, acts as a linear effect on filters with small ensemble sizes. Under weakly-nonlinear error growth, the span of the anomalies projects strongly onto the span of the leading BLVs — therefore, the Riccati equation, Eq. (38), highlights an important, and previously unexplained, mechanism driving the systematic underestimation of the forecast error in ensemble-based Kalman filters. This mechanism likewise explains one reason why, in certain scenarios, covariance inflation has been successful in preventing filter divergence.

The role of inflation we describe differs from previous studies, e.g., the work of Palatella and Trevisan (2015), which studied the nonlinear interactions of error in perfect models. The phenomena of dynamical upwelling is also independent of the mis-estimation of error due to a finite sample size representing the error statistics (Bocquet et al., 2015). Rather, we exhibit an effect which can contribute to filter divergence over short time scales in ensemble data assimilation when the error dynamics are linear or weakly-nonlinear, and uncertainty is forced by additive model errors. This persistent dynamical upwelling of

errors from the unfiltered space into the ensemble subspace is a phenomena which we prove analytically in linear models, and demonstrate numerically to be a valid approximation of weakly-nonlinear error growth in nonlinear models for reduced rank extended Kalman filters.

If we treat the standard EnKF as Monte Carlo estimate of the error statistics characteristic of the KF-AUSE covariance, Eq. (38), the dynamical upwelling explains a significant role for covariance inflation in the EnKF. But our results also suggest

that covariance inflation may potentially be obviated by: (i) sufficiently increasing the ensemble size to include asymptotically stable modes that produce transient instabilities; (ii) increasing the rank of the analysis update itself, with a hybridized gain; (iii) parameterizing the upwelling of error via a hyperprior which targets the evolution of forecast errors; or (iv) some combination of the above. The necessary ensemble size to mitigate the effect of transient instabilities can in principle be studied empirically by examining the local variability of the exponents as in Fig. 3, and their forcing on the evolution of perturbations as in the

numerical study performed by Grudzien et al. (2018) in their section 5. However, computational limits on ensemble sizes in

large, geophysical models, and non-stationarity of the system's dynamics, can limit the effectiveness of this approach. Our understanding of the dynamics of error propagation thus opens new opportunities in algorithm design, where a combination of the above techniques may be used directly to ameliorate the effects of dynamical upwelling, and produce more robust ensemble-based filters.

Where there is dynamical chaos, AUS will continue to be a robust framework for the theory of data assimilation in physical models. Understanding the dynamical mechanisms that govern the evolution of error in fully nonlinear data assimilation, e.g., the unstable-neutral manifolds of a (stochastic) chaotic attractor, will be the subject of future research and may be considered the logical extension of the framework put forward by Anna Trevisan — her insight to the underlying processes in assimilation will continue to provide inspiration to both developers and practitioners of data assimilation methods.

*Acknowledgements.* This work benefited from funding by the project REDDA of the Norwegian Research Council under contract 250711. CEREA is a member of the Institut Pierre-Simon Laplace (IPSL). The authors thank the three anonymous referees, Steve Penny and Patrick Raanes, for their valuable feedback on this work. The authors would like to show their gratitude and respect for Anna Trevisan, and the impact she has had on the understanding of theoretical data assimilation.

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
