# Peer review of "Chaotic dynamics and the role of covariance inflation for reduced rank Kalman filters with model error"

_Nonlinear Processes in Geophysics, 2018_

## Referee Comment (RC1) · Anonymous Referee #1 · 6 Mar 2018

Dear Editor,

I have read the manuscript Chaotic dynamics and the role of covariance inflation for reduced rank Kalman filters with model error by Colin Grudzien, Alberto Carrassi, and Marc Bocquet.

This paper deals with the role of covariance inflation for reduced rank, in particular for AUS variants, of Kalman filters, when the time evolution is affected by model error.

The authors analyze in a very formal and rigorous way the effect of the presence of model error in both filtered and unfiltered space. In particular they emphasize the role of the dynamical upwelling of model errors from the unfiltered to the filtered space and

the importance of this effect in leading to filter divergence.

The difference between covariant and Backward Lyapunov vectors is already known but the authors treat this subtle point in a very precise way and this is surely a merit for the paper.

The authors are also right when they state that this form of divergence is not due to nonlinearity of error dynamics and consequently is different from the case described in Palatella and Trevisan 2015.

the authors of a recent paper

Palatella, Luigi, and Fabio Grasso. "The EKF-AUS-NL algorithm implemented without the linear tangent model and in presence of parametric model error." SoftwareX 7 (2018): 28-33.

show a possible way to manage model error in the framework of EKF-AUS filters in a very low dimensional model. In particulare they suggest that a new direction in the phase-space should be filtered for each degree of freedom of model error. Their approach is obviously unfeasible in high dimensional model, so I think that the approach followed by the authors of the manuscript under examination is important and worth of publication on NPG.

---

## Referee Comment (RC2) · Anonymous Referee #2 · 7 Mar 2018

Inflation has been a necessary evil for ensemble Kalman filters. Originally it was thought to be related primarily to the sampling error. Recent studies, however, showed that nonlinearity, formative/informative hyperpriors by Bocquet et al. 2015, and a reduced rank representation of a covariance matrix have a large influence on the stability of an ensemble Kalman filter. The latter one was thoroughly studied by Anna Trevisan and her collaborators for a perfect model scenario (the authors of the submitted manuscript make a good overview in introduction). A recently submitted, I believe, paper by Grudzien, C., Carrassi, A., and Bocquet, M.: "Asymptotic forecast uncertainty and the unstable subspace in the presence of additive model error" available only as an arXiv preprint arXiv:1707.08334, 2017, studies the influence of a reduced

rank representation of a covariance matrix for an additive model noise. The submitted manuscript is a continuation of the latter work, where the authors derive explicit equations for the error propagation in both filtered and unfiltered subspaces for linear and linearised models. The derivations are simple but have not been done for ensemble Kalman filters up to my knowledge. The manuscript has potentially a merit but needs to undergo major revisions.

Major comments:

1) In numerical experiments with a nonlinear model, I cite the authors: "At each observation time, before observations are given, the true trajectory is perturbed by additive Gaussian noise with a proscribed covariance Q, fixed in time". This set-up is for an additional observational error rather than a model error for a nonlinear case. Instead the "true" solution should be obtained from a stochastic nonlinear model integrated by the Euler-Maruyama scheme, for example.

2) The authors should compare their results with the ensemble Kalman filter with hyperpriors by Bocquet et al. 2015, as the goal of the latter paper was to remove the intrinsic need for inflation.

Minor comments:

1) How was the inflation factor $\alpha$ obtained? What is its value?

2) Additive inflation should be also studied. It is a simple extension which will, however, bring new insights.

3) The authors use complete observations. A study of incomplete observations is again a simple extension which will bring more merit to the manuscript.

Technical comments:

1) Italics is used too often in the text to give an emphasis, it should be avoided.

---

## Short Comment (SC1) · 7 Mar 2018

The following comments are a "hot take" from a quick reading of the manuscript, and do not have the quality of a full review.

From my perspective as an EnKF practitioner, the paper is highly interesting. It derives explicit formulea for "upwelled model error", a source of error whose existence I only vaguely suspected. It also demonstrates how inflation is a workable remedy to counteract this phenomenon.

**Suggested modifications:**

Sometimes "correcting [some of the] BLVs" is used. It should rather be "correcting the uncertainty along the BLVs", right?

Both Prop 1 and Def 3 seem to refer to the exact same set of equations. This is somewhat confusing.

P11 L26 uses "doesn't" which is a no-no

P12 L25. "The dynamical upwelling of model error differs from the sampling errors induced by nonlinear dynamics in perfect models, treated in the modified EKF-AUS-NL (Palatella and Trevisan, 2015) and in the finite size ensemble Kalman filter, (EnKF-N) (Bocquet, 2011; Bocquet et al., 2015)." I'd say EKF-AUS-NL is concerned with the **truncation** error, while the EnKF-N is (mainly) concerned with **sampling error**

P13 L21-22. Insert something like the following sentence: "A survey of the causes for inflation in the EnKF (model error, sampling error, intrinsic bias, localization, non-Gaussianity, and upwelling) is given by [1]."

P21 L7: "the reduced rank" –> "a low-rank"

Discussion could be slightly less scattered. Repetition is good, but sometimes I'm wondering if I have already read this point, or if it's a new one.

P14 L11: ensemble based –> ensemble-based P14 L19: simulates –> emulates

P14 L24: hypothesized –> found evidence that

P19 L7: Just to help lazy readers like me: could you state whether the inflation (through K) affects the mean **and cov** updates, or just the mean.

P20 L2-3: change to: multiplicative inflation will also **need to** compensate for the **truncation** error as described by Palatella and Trevisan (2015),

P20 L6: samples –> "sample points" or "members" or "particles"

**Refs:**

```
{
[1]
@article{raanes2018adaptive,
  title={Adaptive covariance inflation in the ensemble Kalman filter by Gaussian scale mi
  author={Raanes, Patrick N and Bocquet, Marc and Carrassi, Alberto},
  journal={arXiv preprint arXiv:1801.08474},
  year={2018}
}
```

---

## Author Comment (AC1) · 12 May 2018

**Introduction**

The authors would like to express their gratitude for referees' critique of our manuscript. We believe that in formulating our responses, we have developed additional insights to the problem, and its extensions to future work, which we intend to discuss in our revised manuscript. However, before entering into details we would like to reiterate the purpose of this work: we have contributed a rigorous proof of phenomenon, demonstrating one of the underlying mechanisms that determine the role of covariance inflation in reduced rank Kalman filters, in a formulation characteristic of the standard ensemble Kalman filter. We have not, however, made any claim to providing a practical, computationally efficient, means of correcting for this phenomenon. Similar to how we view the seminal work of AUS as a theoretical framework for understanding the properties of ensemble based covariances in the presence of chaotic dynamics (and in the absence of model error), the derivation of KF-AUSE is meant to be used as a theoretical explanation for the empirically observed properties of ensemble based covariances in the presence of chaotic dynamics and additive model errors. This is emphasized already in the original submission throughout sections 3.4 and section 5, and specifically in: (i) lines 5 - 16, page 12; (ii) the discussion in page 13; (iii) lines 8 - 18, page 16; (iv) lines 3 - 5, page 18; (v) lines 14 - 19 page 19; (vi) lines 1 - 4 page 20; (vii) and lines 5 - 13, page 21. It is in the context of the above discussion, in which we have presented our results, that we will respond to the referees' comments.
* * *
**1 Responses to referee 1**

**Comment**(I) ————————————————————————————————————

**Referee:**

"The difference between covariant and Backward Lyapunov vectors is already known but the authors treat this subtle point in a very precise way and this is surely a merit for the paper."

**Response:**

We are very grateful that the referee has appreciated this subtlety, which we wanted to emphasize in lines 26 - 33 of page 9 and lines 1 - 15 of page 10. We believe that, although this is a fine distinction, explaining the equivalences and differences in the span and orthogonal compliments of the two sets of vectors has important consequences in designing filtering techniques used to treat the effects of dynamical upwelling.

**Comment**(II) ———————————————————————————————————

**Referee:**

"The authors of a recent paper Palatella, Luigi, and Fabio Grasso. "The EKF-AUS-NL algorithm implemented with- out the linear tangent model and in presence of parametric model error." SoftwareX 7 (2018): 28-33. show a possible way to manage model error in the framework of EKF-AUS filters in a very low dimensional model. In particular they suggest that a new direction in the phase-space should be filtered for each degree of freedom of model error. Their approach is obviously unfeasible in high dimensional model, so I think that the approach followed by the authors of the manuscript under examination is important and worth of publication on NPG."

**Response:**

We appreciate the referee highlighting this recent publication and we will discuss it in our review of recent literature in the conclusion of our manuscript.
* * *
**2 Responses to referee 2**

**2.1 Major comments**

**Comment**(I) ─────────────────────────────────────────────────

**Referee:**

"'In numerical experiments with a nonlinear model, I cite the authors: 'At each observation time, before observations are given, the true trajectory is perturbed by additive Gaussian noise with a prescribed covariance Q, fixed in time'. This set-up is for an additional observational error rather than a model error for a nonlinear case. Instead the "true" solution should be obtained from a stochastic nonlinear model integrated by the Euler-Maruyama scheme, for example.'"

**Response:**

We apologize for not being sufficiently clear in explaining our nonlinear experimental set up, which is mathematically consistent with the model error scenario for discrete, nonlinear maps. To reiterate, at each observation time, before observations are given, the true trajectory is perturbed (in model space) by additive Gaussian noise with a prescribed covariance $\mathbf{Q}$, fixed in time. Define the nonlinear map $\boldsymbol{\Psi}(t_0, t_1) : \mathbb{R}^n \to \mathbb{R}^n$ be the flow map, generated from the Lorenz-96 equations

$$L^m(\mathbf{x}) = -x^{m-2}x^{m-1} + x^{m-1}x^{m+1} - x^m + F, \tag{1}$$

that takes the model state from time $t_0$ to $t_1$. Then, noting that $\boldsymbol{\Psi}(t, t+\delta) = \boldsymbol{\Psi}(s, s+\delta)$ for all $t$ and $s$, we will define $\boldsymbol{\Psi}_\delta \triangleq \boldsymbol{\Psi}(0, \delta)$. In our experiments, the "truth" is thus evolved via the equation,

$$\mathbf{x}_{k+1} = \boldsymbol{\Psi}_\delta(\mathbf{x}_k) + \mathbf{w}_{k+1}, \qquad\qquad \mathbf{w}_{k+1} \sim N(\mathbf{0}, \mathbf{Q}) \tag{2}$$

while the mean trajectory of the "model" state is given by the deterministic evolution, $\mathbf{x}_{k+1}^{\mathrm{b}} = \boldsymbol{\Psi}_\delta(\mathbf{x}_k^{\mathrm{b}})$. In our experimental design, the extended Kalman filter estimates the state of the non-linear "true" state, perturbed by the noise $\mathbf{w}_k$, Eq. (2), and $\mathbf{M}_k$ (the linear propagator for the covariance forward evolution) is derived by the map $\nabla \boldsymbol{\Psi}_\delta\big|_{\mathbf{x}_k^{\mathrm{b}}}$. This experimental configuration is mathematically consistent with the extended Kalman filter for a discrete nonlinear map with model error, and is a standard formulation for model error twin experiments, utilized by e.g, Mitchell and Carrassi (2015); Sakov et al. (2018), with the configuration using the circulant covariance matrix, $\mathbf{Q}$, drawn specifically from Raanes et al. (2015). The interval between observations $\delta$ controls the nonlinearity of the map, where our chosen configuration can be considered weakly-nonlinear. We will include the above expanded discussion in our revision.

Regarding the use of stochastic differential equations (SDEs), we supply these simulations here in our response, but we decline from including these results in the revision. In particular, we do not believe they add substantial additional value to our manuscript as:

- the results are almost identical to those derived from the discrete EKF configuration;
- their presentation requires significant additional explanation, as many readers are unfamiliar with mathematically robust simulations of SDEs;
- there is not as simple an interpretation of the local Lyapunov exponents for an SDE system as in the case of the discrete map perturbed by noise.

We elaborate on the above points in the following, where we will describe the configuration of our SDE simulations and the derived results.

Let

$$\mathrm{d}\mathbf{x} = \mathbf{L}(\mathbf{x})\mathrm{d}t + \sigma\mathrm{d}\mathbf{W}(t) \tag{3}$$

where $\mathbf{L}$ is defined in Eq. (1), $\mathbf{W}(t)$ is an $n$-dimensional, standard normal Weiner process, and $\sigma > 0$ is a diffusion coefficient, representing uniform variances of the noise in space and time. We note that for SDEs with additive noise (the above configuration being a special case), there is no difference between the Itô and Stratonovich integral of the SDE (Kloeden and Platen, 2013, see page 109), which simplifies our discussion. We utilize the differential operators defined on page 339, and the approximations for the multiple Stratonovich integrals on pages 202 - 203, to derive the integration rule for the order 2.0 strong Taylor scheme on page 359 of Kloeden and Platen (2013). The order 2.0 strong Taylor scheme reduces to the usual order 2.0 Taylor scheme in a deterministic setting, and the mean trajectory of the "model" state is propagated with the deterministic, order 2.0 Taylor scheme. The time step for both the true and model trajectory is fixed at $h = 0.0025$. The tangent-linear equations of the "model" trajectory is integrated with an order 4.0 Runge-Kutta scheme, with time step $0.005$. The interval between observations is kept fixed as $\delta = 0.1$, maintaining the weakly-nonlinear error growth.

We choose the diffusion coefficient $\sigma = 0.25$, plotting the analysis RMSE of EKF, EKF-AUS and EKF-AUSE over 100,000 forecast cycles. In the case of $\sigma = 0.25$, the results are almost identical to Fig. 2 of our original manuscript. We find that diffusion coefficients of $\sigma = 0.1$ and $0.5$ are qualitatively the same and are not pictured here.

[Figure]

**Figure 1.** SDE diffusion $\sigma = 0.25$. Analysis RMSE of EKF-AUS plotted with triangles and EKF-AUSE plotted with X's, varying over the rank of the sub-optimal gain. Horizontal lines are the observational error standard deviation and EKF analysis RMSE. Note the log scale of the $y$-axis.

Given the similarity of the SDE simulation with diffusion coefficient of $\sigma = 0.25$ (Fig. 1 above) to our earlier simulation with discrete nonlinear maps, we choose this parameter configuration to evaluate the impact of multiplicative inflation on the reduced rank EKF-AUS. We, once again, choose a filtered subspace of dimension 17 and vary the inflation parameter $\alpha$ on the x-axis in Fig. 2 below.

[Figure]

[Figure]

**Figure 2.** SDE diffusion $\sigma = 0.25$. Analysis RMSE of EKF-AUS (y-axis), correction rank 17, with multiplicative inflation plotted versus the inflation value $\alpha$ (x-axis). Horizontal lines are the observational error standard deviation, EKF-AUSE and EKF analysis RMSE. Note the log scale of the $y$-axis.

For the diffusion coefficient of $\sigma = 0.25$, the results for the SDE experiment are almost identical to those of our earlier experiments with discrete nonlinear maps.

Due to the similarity of the results, and the required addition explanation of the experimental configuration for SDEs, we do not believe that it is justified to include both the discrete map and SDE experimental configurations. Given a choice between the two designs, we prefer to use the discrete nonlinear map configuration, as in this case, there is an easy to interpret role of the local Lyapunov exponents which is more difficult to define in the case of an SDE, and goes beyond the scope of this work. We will, however, remark that: (i) the results are qualitatively the same in the SDE configuration; (ii) however, the full extension of AUS techniques to the presence of stochastic differential equations goes beyond the scope of this work and will be the subject of future research.

**Comment**(II) _______________________________________________________________

**Referee:**

"The authors should compare their results with the ensemble Kalman filter with hyperpriors by Bocquet et al. 2015, as the goal of the latter paper was to remove the intrinsic need for inflation."

**Response:**

In discussing a comparison between the EnKF-N and the ideal recursion represented in EKF-AUSE, please note the following: the original EnKF-N (Bocquet, 2011; Bocquet et al., 2015) was designed to be used in the absence of model errors, in order to treat the misrepresentation of the statistics of the EnKF due to sampling errors. The construction for the EnKF-N, moreover, utilizes the hypothesis that the effective uncertainty lies within the span of a reduced rank ensemble. In the case of a perfect model with weakly nonlinear error evolution, this is a well posed hypothesis as evidenced by the results of Gurumoorthy et al. (2017); Bocquet et al. (2017). In this case, we can consider the forecast error evolution of an ideal, reduced rank Kalman filter to be asymptotically equivalent to the forecast error evolution of the true Kalman filter. Specifically, it is demonstrated that errors in the span of the trailing, stable BLVs vanish exponentially, and the EnKF-N does not need to treat the persistent upwelling of uncertainty that is present in the case of model errors. The EnKF-N of (Bocquet, 2011; Bocquet et al., 2015), rather seeks to address the sampling errors in ensemble based Kalman filters, especially in the presence of nonlinearity, which constitutes a wholly different source of error and reason for inflation.

Therefore, comparing the EnKF-N of (Bocquet, 2011; Bocquet et al., 2015) with EKF-AUSE would not provide any meaningful conclusions, and would conflate the disparate sources of uncertainty, as we already discussed throughout section 3.4, and lines 1 - 7, page 20, of our manuscript. Indeed, the recent work of Raanes et al. (2018), providing an extension of the EnKF-N to the presence of model errors, utilizes an additional, adaptive inflation factor to account for the underestimation of uncertainty due to model errors. However, in our manuscript we have emphasized that although the EnKF-N does not currently take into account dynamical upwelling in its formulation to treat the presence of model errors, an eventual goal would be to incorporate the ideal recursion for a reduced rank filter into the hyperprior. This is discussed specifically in: lines 21 - 23, page 13; lines 28 - 31, page 13; lines 8 - 15, page 16; lines 7 - 11, page 21. Formerly, the hyperprior of the EnKF-N has been uninformative in the sense that the hyperprior on the covariance is with respect to all positive semi-definite matrices, thus constituting a Jefferys prior. However, as demonstrated in Fig. 1 of our manuscript, for a reduced rank filter in the presence of model error, there is additional structure which gives a refinement to this set of matrices. Specifically, if the EnKF-N has a reduced rank filtered subspace, then we may view the EnKF-N as a Monte Carlo estimate of the ideal recursion of KF-AUSE, with an error covariance that is stratified across the unfiltered

and filtered subspaces — this is discussed in the manuscript in lines 14 - 18 page 6, and lines 5 - 7, page 21. This work goes beyond the scope of the manuscript and is the subject of future research.

In response to your suggestion, we will expand on our earlier discussions, including reference specifically to the recent submission of Raanes et al. (2018), and further clarify the differences between the two treated sources of uncertainty.

**2.2 Minor comments**

**Comment**(I) ——————————————————————————————————————————————

**Referee:**

"How was the inflation factor $\alpha$ obtained? What is its value?"

**Response:**

In our submission, page 19, lines 11-12 we state,

"Additionally, we plot the analysis RMSE of EKF-AUS as a function of the inflation value applied to the forecast error covariance, with the inflation values plotted as triangles."

We apologize that this sentence was not totally clear. We meant to indicate that the selected inflation is equal to the x-value at each point marked with a triangle in the graph, with the corresponding y-value equal to the RMSE. In our revisions we will indicate that the values of inflation, $\alpha$, are given as the x-values in the graph, for evenly spaced points in $[1, 4]$ at increments of $0.1$.

**Comment**(II) —————————————————————————————————————————————

**Referee:**

"Additive inflation should be also studied. It is a simple extension which will, however, bring new insights."

**Response:**

We believe that it is interesting, and highly relevant, to study the effect of covariance and/or gain augmentation to reduce the effect of the dynamical upwelling and the presence of residual error in the unfiltered directions. We earlier summarized our thoughts on additive inflation in our original submission in lines 1 - 34, page 13, and lines 15 - 18, page 16. In these sections, we emphasized that augmenting a reduced rank gain by additive inflation or hybridization may reduce the effect of dynamical upwelling by keeping errors in the trailing BLVs small. However, we also state that this will generally induce sampling errors by corrupting the error estimates in the standard KF recursion. This will likewise induce mis-estimation of the error in KF-AUSE, which is simply the analytically derived forecast error in the case of a reduced rank gain.

The logical extension of our work studying additive inflation would thus include deriving the ideal recursion on the forecast error covariance with respect to an ensemble based gain, augmented with a sub-optimal correction in the trailing BLVs. By deriving the recursion, one can analytically study

the effects of the sub-optimal correction on the propagation of errors, and how various computationally efficient approximations of this error evolution affects the RMSE. This would be the exact analogue of the work that we have completed, where we have studied the forecast error evolution with respect to a reduced rank gain, and the approximation of the dynamical upwelling in the ideal recursion with the computationally efficient alternative of multiplicative inflation. However, the mathematical complexity in obtaining an ideal recursion for additive inflation, as described above, is such that it cannot be included in this manuscript.

On the other hand, we may treat the sources of uncertainty described in this work approximately. We have highlighted this possibility, proposing a combination of some form of gain augmentation, with a hyperprior to account for the corrupted error estimates, to target these sources of uncertainty — this is suggested in lines 28 - 32, page 13, lines 8 - 18, page 16, lines 7 - 11, page 21. However, the purpose of this manuscript is only to provide a rigorous proof of phenomenon, and introducing the above approximations goes beyond the scope of this work. In order to more fully explain the significance of these extensions to additive inflation, and its mathematical complexity, we will include an additional discussion section in our revised manuscript elaborating on the above points.

**Comment**(III)

"The authors use complete observations. A study of incomplete observations is again a simple extension which will bring more merit to the manuscript."

**Response:**

We agree that this is a simple extension, and as such we provide a numerical demonstration in this response. Specifically, using our original configuration of discrete, nonlinear maps with additive noise, we simulate the effect of reducing the dimension of the observational subspace while keeping all other parameters fixed. However, we do not believe that the results with reduced observations: (i) are qualitatively different from the results with a fully observed system, or (ii) add significant new information about the effect of the dynamical upwelling in a reduced rank Kalman filter. The major difference in the results with reduced observations lies only in the minimum rank of the filtered subspace to prevent filter divergence.

[Figure]

**Figure 3.** EKF-AUS and EKF-AUSE RMSE, plotted versus the rank of the filtered subspace. Observations are taken at all odd nodes $\mathbf{x}_k^i$ for $i \in 1, \cdots, 39$.

We see once again that EKF-AUSE has a lower minimum, and in general lower RMSE, than EKF-AUS. In the case of an observational subspace of dimension 20, Fig. 3, the minimum rank of the filtered subspace to prevent divergence is 20 for EKF-AUSE, while EKF-AUS has a minimum rank of 26. For all RMSE values not pictured in Figs. 3, the EKF-AUSE and EKF-AUS diverge due to numerical instability. We find qualitatively similar results when using an observational dimension of $d = 30$, and these results are not pictured here.

We decline from including these results in our revised manuscript, though, when we discuss the qualitative similarity of other experimental configurations, we will discuss that when reducing the observational dimension, the usual pattern persists. This will be added to the discussion in our original submission, in line 18, page 17, through line 3, page 18.

**Comment**(IV)

"Italics is used too often in the text to give an emphasis, it should be avoided."

**Response:**

We apologize for this distraction. We have removed most, but not all, of the italics. We have chosen to use the emphasis more selectively in a few key spots to emphasize important points — we hope that this is more satisfactory.

**3   Response to short comments**

Because the short comments are on relatively minor points, we will conclude here by saying that we appreciate the feedback and will implement the suggestions. Most importantly, we separate the definition of the KF-AUSE Riccati equation, and the related proposition, so that we can state the proposition in its fullest generality — this will be included in the revised text.

**References**

Bocquet, M.: Ensemble Kalman filtering without the intrinsic need for inflation, Nonlinear Processes in Geophysics, 18, 735–750, 2011.

Bocquet, M., Raanes, P., and Hannart, A.: Expanding the validity of the ensemble Kalman filter without the intrinsic need for inflation, Nonlinear Processes in Geophysics, 22, 645, 2015.

5   Bocquet, M., Gurumoorthy, K., Apte, A., Carrassi, A., Grudzien, C., and Jones, C.: Degenerate Kalman Filter Error Covariances and Their Convergence onto the Unstable Subspace, SIAM/ASA Journal on Uncertainty Quantification, 5, 304–333, 2017.

Gurumoorthy, K., Grudzien, C., Apte, A., Carrassi, A., and Jones, C.: Rank deficiency of Kalman error covariance matrices in linear time-varying system with deterministic evolution, SIAM Journal on Control and Optimization, 55, 741–759, 2017.

Kloeden, P. and Platen, E.: Numerical Solution of Stochastic Differential Equations, Stochastic Modelling and Applied Probability, Springer
10   Berlin Heidelberg, 2013.

Mitchell, L. and Carrassi, A.: Accounting for model error due to unresolved scales within ensemble Kalman filtering, Quarterly Journal of the Royal Meteorological Society, 141, 1417–1428, 2015.

Raanes, P., Carrassi, A., and Bertino, L.: Extending the square root method to account for additive forecast noise in ensemble methods, Monthly Weather Review, 143, 3857–3873, 2015.

15   Raanes, P., Bocquet, M., and Carrassi, A.: Adaptive covariance inflation in the ensemble Kalman filter by Gaussian scale mixtures, arXiv preprint arXiv:1801.08474, 2018.

Sakov, P., Haussaire, J., and Bocquet, M.: An iterative ensemble Kalman filter in presence of additive model error, Quarterly Journal of the Royal Meteorological Society, 2018.

---

## Author Response (AR2)

**1 Introduction**

The authors would like to express their gratitude for reviewers' critique of our manuscript, especially in terms of their support in refining the consistency of the narrative. We believe that in responding to these reviews, we have greatly improved the clarity and consistency of our exposition and, therefore, the impact of the work.

- 5 In response to the comments of the referees, we believe that we have been unclear in discussing our position on multiplicative inflation. We want to emphasize that while we believe that dynamical upwelling explains one of the basic reasons why multiplicative inflation has been successful in preventing filter divergence in certain model error scenarios, we also recognize that there are superior approaches that can include a more direct parametrization of the upwelling, rather than its emulation via a scalar coefficient. We earlier pointed to the possibility of using less ad hoc approaches to account for dynamical upwelling
- 10 by using, e.g., hyperpriors and hybridization to target this source of error directly. However, we agree with the referees that we had earlier overstated the connection to inflation and that this overstatement distracted from our conclusions. We have removed the statement of the "intrinsic role of inflation" throughout the work, the motivation for which was indeed

only made empirically, not mathematically. In our revised text, we have now focused on how the main results (demonstrating dynamical upwelling and the underestimation of forecast error in the ensemble span with the reduced rank Kalman filter recursion) provides an additional theoretical explanation on how covariance inflation has been successful, in certain scenarios,

15 recursion) provides an additional theoretical explanation on how covariance inflation has been successful, in certain scenarios, in preventing filter divergence. Particularly, we have included further exposition on how the terms in the KF-AUSE Riccati equation can be interpreted as providing an inflation factor in the ensemble span, over the standard EnKF recursion. While we do not provide a formal proof, we believe that this better motivates the above conclusions. We also further emphasize how more sophisticated approaches including gain augmentation are likely better suited to deal with unfiltered errors in the trailing

**20 BLVs and its resultant upwelling.**

In the following sections we will respond to the referees point by point.

**2 Responses to referee 2**

**25**

30

35

40

Comment(I) \_

**Referee:**

"I agree with the authors that 'multiplicative inflation (in the ensemble span) neglects the fundamental issue that the unfiltered error lying outside of the ensemble span can be the major driver of the uncertainty in a reduced rank filter with model error'. This could be seen from eq. (38) and eq. (39). However, it should be studied analytically rather than numerically how these missing terms are linked to the inflation, meaning that inflation should be derived analytically rather than empirically, since the authors did develop all the mathematical tools."

**Response:**

We thank the referee for this comment. While we do not have a true mathematical proof of the connection to inflation, we believe that the terms (37b) - (37d) in equation (37) of the revised draft support the notion of an empirical inflation term, present in KF-AUSE Riccati equation. Specifically, term (37a) represents the standard reduced rank recursion for the error in the ensemble span in the generic reduced rank EnKF. Therefore, the remaining terms in the exact recursion, (37b) - (37d), can be considered an empirical inflation term missing in the standard recursion.

Particularly, these terms (37b) - (37d) are explicitly determined by the unfiltered error and its cross covariance with the filtered error. In response to the referee, we have further discussed the effect of the missing terms (37b) - (37d) and their interpretation as empirically inflating the error estimate of the standard EnKF recursion. This additional exposition is contained in section 3.4.

**5 Comment(II) \_\_\_\_**

10

15

20

| Referee:                                                                                                                                                                                                                                                                                                                                                                                                                                      |
|-----------------------------------------------------------------------------------------------------------------------------------------------------------------------------------------------------------------------------------------------------------------------------------------------------------------------------------------------------------------------------------------------------------------------------------------------|
| "Moreover, the empirical inflation parameter is chosen to be between 1 and 4, which means the covariance was inflated 4 times, while typically it is inflated by 20% at most. Apart from that I am surprised that EKF-AUS performs equivalently for inflation between 1.5 and 4 (only slightly worse for high inflation). I would expect that the RMSE substantially increases for higher inflation."                                         |
| Response:                                                                                                                                                                                                                                                                                                                                                                                                                                     |
| We thank the referee for noting this point. We believe that this is indeed the correct interpretation
for operational conditions where observation errors are more significant and observations are typ-
ically sparse. However, in our twin experiment configuration with a completely observed system
and with Gaussian observation error, with uniform, homogeneous covariance $\mathbf{R}_k \triangleq 0.25\mathbf{I}_{40}$ , we |

is indeed the correct interpretation ignificant and observations are typwith a completely observed system eous covariance  $\mathbf{R}_k \triangleq 0.25 \mathbf{I}_{40}$ , we believe that the numerical results are consistent. While, the performance of the reduced rank EKF-AUS with inflation doesn't degrade significantly with high inflation versus the optimally tuned one, we believe this is to be expected as the filtering error becomes dominated by the relatively precise and dense observations. Moreover, while the degradation of the RMSE of EKF-AUS is not significant with respect to the optimally tuned inflation value, the degradation is significant with respect to the full rank extended Kalman filter.

**Responses to Steve Penny** 3 25**

We thank Steve Penny for his openness and candor in the review process. We address his comments directly below.

**Comment**(I)

30

**Referee:**

"...I think it should be made very clear at the start that this is a simple subset of the full range of possible model error types, e.g. around page 2, line 12."

35

**Response:**

We thank the referee for this comment, as it indeed clarifies the text. We have now included additional exposition in the introduction section clarifying that this work is concerned with a simplified assumption on the type of model errors.

5

10

15

20

25

30

35

40

**Referee:**

"...I agree with the previous reviewer in that there is a bit of a jump from section 3 into the results. I was expecting a bit more mathematical development to show exactly why multiplicative inflation is the choice made by the authors to address the problem of dynamic upwelling. As a reader, the motivation is not clear. I would appreciate at least one more subsection at the end of section 3 walking the reader through the motivation and justification for turning to multiplicative inflation to resolve the issue of dynamic upwelling."

**Response:**

We thank both the referees for pushing us to clarify this exposition, which we believe was formerly lacking. Section 3.4 in our earlier draft attempted to motivate our numerical study, and the connection between the KF-AUSE Riccati equation and multiplicative covariance inflation. As is discussed in our response to referee 2, comment (I) above, we lack a formal mathematical proof of the process of upwelling leading to an exact inflation term. However, we believe that there is sufficient analytical justification from the terms of equation (37) to motivate the use of the KF-AUSE Riccati equation as a theoretical interpretation of covariance inflation. To re-iterate, the terms (37b) - (37d) are lacking in the estimated forecast error in the ensemble span in the standard EnKF recursion. Particularly, the term (37a) represents the usual estimated forecast error recursion for the EnKF. The introduction of the terms (37b) - (37d) for the recursion of the filtered error in the exact KF-AUSE Riccati equation motivates the interpretation that upwelling empirically inflates the true uncertainty in the ensemble span versus the estimated uncertainty of the EnKF. Rather than including an additional section after 3.4, we have introduced a longer exposition on this point, clarifying the role of the terms (37b) - (37d) in the KF-AUSE Riccati equation as an empirical inflation to the estimated error for the EnKF.

**Comment(III) .**

**Referee:**

"I am generally of the opinion that multiplicative inflation is a simple but inappropriate tool for use in general ensemble-based data assimilation. I am aware of the original motivation for its origins, but my experience has shown in many circumstances that in trying to correct one problem (e.g. an under-dispersive ensemble spread due to under-sampling or model error), it has the potential to lead to catastrophic filter divergence. I am open to being convinced to the contrary with a rigorous justification, but what I have seen so far has not quite succeeded."

**Response:**

We thank the referee for this comment and their interest in our derivations. We wish to emphasize, however, that our conclusion is that the KF-AUSE Riccati equation provides a novel theoretical interpretation of how multiplicative covariance inflation has been successful, in certain scenarios, in preventing filter divergence. However, we do not wish to convince the reader that multiplicative inflation is the preferred option to account for the missing terms in the recursion of the filtered errors, (37b) - (37d). We conclude that there are likely more appropriate methods of accounting

for the dynamic upwelling phenomenon in a reduced rank filter, as evidence by our results, and we suggest that hyperpriors, hybridization, or some combination therein would be a preferable approach.

|    | Comment (IV) |                                                                                                                                                                                                                                                                                                                                                                                                                                                                                                                                                                                                                                         |
|----|---------------------|-----------------------------------------------------------------------------------------------------------------------------------------------------------------------------------------------------------------------------------------------------------------------------------------------------------------------------------------------------------------------------------------------------------------------------------------------------------------------------------------------------------------------------------------------------------------------------------------------------------------------------------------|
| 5  |                     | Referee:                                                                                                                                                                                                                                                                                                                                                                                                                                                                                                                                                                                                                                |
|    |                     | "if any statement about the validity of multiplicative inflation method is to be made, then it should
have a rigorous analytical justification. However, I believe the authors may be very close to that
goal for a specific subset of problem relating to the case of additive model errors. As long as the
authors make this limitation clear from the beginning, then I believe the work should be published."                                                                                                                                                                                                              |
| 10 |                     |                                                                                                                                                                                                                                                                                                                                                                                                                                                                                                                                                                                                                                         |
|    |                     | Response:                                                                                                                                                                                                                                                                                                                                                                                                                                                                                                                                                                                                                               |
| 15 |                     | We are very appreciative of the referee's comment. We believe that it has clarified an important
point that, while covariance inflation may be successful in certain scenarios in dealing with the
effects of upwelling, our derivations are only applicable to a limited subset of model error and
filtering regimes. In response, we have emphasized throughout the text that the KF-AUSE Riccati
equation provides a theoretical interpretation for the limited success of covariance inflation, but we
see less ad hoc approaches that deal with the phenomenon of dynamical upwelling directly to be
preferable. |
| 20 | Comment (V)  |                                                                                                                                                                                                                                                                                                                                                                                                                                                                                                                                                                                                                                         |
|    |                     | Referee:                                                                                                                                                                                                                                                                                                                                                                                                                                                                                                                                                                                                                                |
|    |                     | " 'The interval between observations $\delta$ controls the nonlinearity of the map'                                                                                                                                                                                                                                                                                                                                                                                                                                                                                                                                                     |
| 25 |                     | This is not correct. It is the interval between random impulsive forcing that controls the nonlin-
earity of the map. You happen to have equal matching intervals between the observing time and
the random forcing, but these need not be identical. The interval between observations is only rel-
evant to the dynamics if you are referring to the dynamics of the combined forecast/analysis data
assimilation system."                                                                                                                                                                                                |
| 30 |                     |                                                                                                                                                                                                                                                                                                                                                                                                                                                                                                                                                                                                                                         |
|    |                     | Response:                                                                                                                                                                                                                                                                                                                                                                                                                                                                                                                                                                                                                               |
|    |                     | We agree with the referee and thank them for their comment. In response, we have re-written our exposition in this section to correct this mistake by stating that,                                                                                                                                                                                                                                                                                                                                                                                                                                                                     |
| 35 |                     | "At each observation time, before observations are given, the true trajectory is perturbed (in model space) by additive Gaussian noise with a prescribed covariance $\mathbf{Q}$ , fixed in time. In general, the random model noise can be injected at different intervals than the interval between observations, affecting the nonlinearity of the error evolution. However, we fix these intervals to be equal for simplicity"                                                                                                                                                                                                      |
| 40 |                     | and later state that                                                                                                                                                                                                                                                                                                                                                                                                                                                                                                                                                                                                                    |
|    |                     | and more state that                                                                                                                                                                                                                                                                                                                                                                                                                                                                                                                                                                                                                     |

"The interval between observations  $\delta$  controls the nonlinearity of evolution of the forecast errors in the combined forecast/analysis data assimilation cycle. Our chosen configuration for the observation interval, and the interval of random forcing in the model, can be considered weakly-nonlinear."

|    | Comment (VI) |                                                                                                                                                                                                                                                                                                                                                                                             |
|----|---------------------|---------------------------------------------------------------------------------------------------------------------------------------------------------------------------------------------------------------------------------------------------------------------------------------------------------------------------------------------------------------------------------------------|
| 5  |                     | Referee:                                                                                                                                                                                                                                                                                                                                                                                    |
|    |                     | "Response, page 5: 'as demonstrated in Fig. 1 of our manuscript, for a reduced rank filter in the presence of model error, there is additional structure which gives a refinement to this set of matrices."                                                                                                                                                                                 |
| 10 |                     | This may be true, but it appears that the additional structure must be dependent on the choice of the presumed model error covariance Q. Is this correct? "                                                                                                                                                                                                                                 |
|    |                     | Response:                                                                                                                                                                                                                                                                                                                                                                                   |
| 15 |                     | We thank the referee for this comment. We completely agree that the choice of $\mathbf{Q}$ will impact the shape of the reduced rank Kalman filter forecast error covariance. In response, we have stated this qualification when we introduce Fig. 6 in the updated section 5.                                                                                                             |
|    | Comment(VII)        |                                                                                                                                                                                                                                                                                                                                                                                             |
|    |                     | Referee:                                                                                                                                                                                                                                                                                                                                                                                    |
| 20 |                     | "Response, page 7: 'major difference in the results with reduced observations lies only in the min-
imum rank of the filtered subspace to prevent filter divergence'                                                                                                                                                                                                                     |
| 25 |                     | Again, I believe that making the distinction between the analysis update interval, observing interval, and impulsive model forcing interval should help to clarify some of the issues. In this case, reducing the observations can change the effective forecast window in a 'staggered' sense, so that any point in the domain has a different time since last having an analysis update." |
|    |                     | Response:                                                                                                                                                                                                                                                                                                                                                                                   |
| 30 |                     | We thank the referee for clarifying this subtlety, which we believe is a more accurate interpretation of the results than the one provided in our earlier response.                                                                                                                                                                                                                         |
|    | Comment(VIII)       |                                                                                                                                                                                                                                                                                                                                                                                             |
|    |                     | Referee:                                                                                                                                                                                                                                                                                                                                                                                    |
| 35 |                     | "I agree with the reviewer, the use of italics for emphasis is unnecessary. My general opinion regarding a scientific or technical paper is that if something is unnecessary then it should not be included."                                                                                                                                                                               |
|    |                     |                                                                                                                                                                                                                                                                                                                                                                                             |

**Response:**

As requested by the referees, we have removed all remaining use of italics for the purpose of emphasis.

|     | Comment (IX) |                                                                                                                                                                                                                                                                                                                                                                                                                                                                                                                                                                                                                                                                                                                                                                                                                                                                                                                                                   |
|-----|---------------------|---------------------------------------------------------------------------------------------------------------------------------------------------------------------------------------------------------------------------------------------------------------------------------------------------------------------------------------------------------------------------------------------------------------------------------------------------------------------------------------------------------------------------------------------------------------------------------------------------------------------------------------------------------------------------------------------------------------------------------------------------------------------------------------------------------------------------------------------------------------------------------------------------------------------------------------------------|
|     |                     | Referee:                                                                                                                                                                                                                                                                                                                                                                                                                                                                                                                                                                                                                                                                                                                                                                                                                                                                                                                                          |
| 5   |                     | "Definition 1 should be reworded so that the description of the indices k and i are not intertwined, e.g.
"Define the matrix Ek to be the orthogonal matrix at time k whose i-th column is the i-th backward Lyapunov vector (BLV), corresponding to the Lyapunov exponent $\lambda_i$ ."                                                                                                                                                                                                                                                                                                                                                                                                                                                                                                                                                                                                                                                      |
| 4.0 |                     |                                                                                                                                                                                                                                                                                                                                                                                                                                                                                                                                                                                                                                                                                                                                                                                                                                                                                                                                                   |
| 10  |                     | Response:                                                                                                                                                                                                                                                                                                                                                                                                                                                                                                                                                                                                                                                                                                                                                                                                                                                                                                                                         |
|     |                     | We thank the referee for their comment. We believe their suggestion clarifies the sentence and we have changed it to the one written above.                                                                                                                                                                                                                                                                                                                                                                                                                                                                                                                                                                                                                                                                                                                                                                                                       |
|     | Comment (X)  |                                                                                                                                                                                                                                                                                                                                                                                                                                                                                                                                                                                                                                                                                                                                                                                                                                                                                                                                                   |
| 15  |                     | Referee:                                                                                                                                                                                                                                                                                                                                                                                                                                                                                                                                                                                                                                                                                                                                                                                                                                                                                                                                          |
|     |                     | "Page 5, lines 1-8, section 2.1:
I'm not sure that 'projection' is the correct term to use here, since $\mathbf{E}^{T}$ itself is not a projection
operator. Perhaps you can alter the terminology slightly to be more precise."                                                                                                                                                                                                                                                                                                                                                                                                                                                                                                                                                                                                                                                                                                            |
| 20  |                     |                                                                                                                                                                                                                                                                                                                                                                                                                                                                                                                                                                                                                                                                                                                                                                                                                                                                                                                                                   |
|     |                     | Response:                                                                                                                                                                                                                                                                                                                                                                                                                                                                                                                                                                                                                                                                                                                                                                                                                                                                                                                                         |
| 25  |                     | To clarify, we have not stated that $\mathbf{E}^{T}$ is a projection. Rather, we believe that we have correctly stated that the operator $\mathbf{E}_{k}^{T}$ takes the vector $\mathbf{z}_{k}$ into its projection coefficients with respect to the orthogonal basis of BLVs. Explicitly, $\mathbf{E}_{k}\mathbf{E}_{k}^{T}$ is an orthogonal projection operator due to the orthogonality of the matrix $\mathbf{E}_{k}$ . In particular, $\mathbf{E}_{k}^{T}\mathbf{v}_{k}$ can be seen as a vector of projection coefficients — applying $\mathbf{E}_{k}$ to the vector $\mathbf{E}_{k}^{T}\mathbf{v}_{k}$ recovers the orthogonal projection of $\mathbf{v}_{k}$ in the span of the BLVs. The same reasoning applies to any sub-slice of the columns of the orthogonal matrix $\mathbf{E}_{k}$ , but in this case we recover the projection coefficients of the vector with respect to the projection into the span of a subset of the BLVs. |
| 30  | Comment (XI) |                                                                                                                                                                                                                                                                                                                                                                                                                                                                                                                                                                                                                                                                                                                                                                                                                                                                                                                                                   |
|     |                     | Referee:                                                                                                                                                                                                                                                                                                                                                                                                                                                                                                                                                                                                                                                                                                                                                                                                                                                                                                                                          |
|     |                     | "Equation (11) is missing a superscript "b" on the first Hx term, otherwise it appears incorrect.
And then based on equation (7) the vk term should be positive, and the He term negative, which I think then corresponds correctly with equation (14) below."                                                                                                                                                                                                                                                                                                                                                                                                                                                                                                                                                                                                                                                                                 |

**Response:**

35

We thank the reviewer for noting this mistake. This issue could have been quite confusing for the readers and we are grateful for the careful read. We have changed the text as described above.

**Referee:**

"Line 20:

The term vk should be known if it is the difference between the observation and the forecast mean."

5

10

**Response:**

 $\boldsymbol{\epsilon}_k$

In our corrected definitions of quantities in the revised manuscript,

 $\mathbf{y}_k \triangleq \mathbf{H}_k \mathbf{x}_k + \mathbf{v}_k,\tag{1}$

$$\stackrel{\Delta}{=} \mathbf{x}_{k}^{\mathrm{b}} - \mathbf{x}_{k},\tag{2}$$

$$\boldsymbol{\delta}_{k} \triangleq \mathbf{y}_{k} - \mathbf{H}_{k} \mathbf{x}_{k}^{\mathrm{b}} = \mathbf{v}_{k} - \mathbf{H}_{k} \boldsymbol{\epsilon}_{k}, \tag{3}$$

knowledge of the innovation process doesn't reveal the realizations of the observational error process. In this case, we have  $\mathbf{v}_k = \mathbf{y}_k - \mathbf{H}_k \mathbf{x}_k$  where the particular realization of the random variable  $\mathbf{x}_k$  is unknown, and generally  $\mathbf{x}_k \neq \mathbf{x}_k^{\text{b}}$ .

**15 Comment(XIII)**

**Referee:**

"I follow equation (17) as long as E is orthonormal. Is that the case?"

**20**

25

30

35

**Response:**

It was stated in the definition of  $\mathbf{E}_k$  that it is assumed be an orthogonal matrix.

Comment(XIV) \_\_\_\_

"Page 11, line 4: Should this be K^?"

**Response:**

We thank the referee for their comment. We have clarified this sentence, that we refer to the reduced rank gain, with update restricted to the span of  $\mathbf{E}_{k}^{f}$ .

Comment(XV) \_

**Referee:**

| "Line 34:                                                                                         |
|---------------------------------------------------------------------------------------------------|
| 'Nevertheless, this design is artificial and would lead to poor filter performance'               |
| This is an absolute statement that would require significantly more evidence in order to state it |
| here. I suggest removing the assertion."                                                          |

**Response:**

We thank the referee for their comment, as we indeed made an overstatement in this sentence. In response we have removed the above sentence, and have reduced our claim to, "These results suggest that it is preferable that the unfiltered space is equal to the span of the trailing BLVs, or equivalently, that the filtered space is defined equal to the span of the leading covariant/ backward Lyapunov vectors."

**Comment(XVI)**

| - | n |
|---|---|
|   | U |
|   | - |

5

"Page 15 (line 32) -16 (line 9)

I like some of the discussion here that was removed describing how previous authors have viewed the use of inflation. It would be nice to put this last paragraph back into the manuscript."

15

25

35

40

**Response:**

**Referee:**

We thank the referee for their comment — in response, we have included the exposition from these lines. However, we have moved this exposition into the discussion in section 5, as we believe it makes the narrative more consistent.

**20 Comment(XVII)**

**Referee:**

"P. 27, Line 10:

'(i) sufficiently increasing the ensemble size to include asymptotically stable modes that produce transient instabilities'

Do you have a proof that the stable modes with larger Lyapunov exponents are more likely to produce the 'dynamic upwelling' than the stable modes with smaller LEs? If not, then modestly increasing the ensemble size may not be a guaranteed solution. "

**30 **Response:**

We thank the referee for this comment, as the limits of this approach were not stated clearly. We have clarified what we intended by this statement, with the following updated text in the conclusion:

[revised manuscript text omitted]

We now consider the covariance of the forecast error in the filtered variables. Using the identity in Eq. (35) we derive the recursion for the filtered error covariance  $\widehat{\mathbf{B}}_{k+1}^{\text{ff}}$  as

$$\mathbf{B}_{k+1}^{\mathrm{ff}} = \mathbf{U}_{k+1}^{\mathrm{ff}} \boldsymbol{\Sigma}_{k} \left( \mathbf{U}_{k+1}^{\mathrm{ff}} \right)^{\mathrm{T}} + \widehat{\mathbf{Q}}_{k+1}^{\mathrm{ff}}$$
(37a)

$$+ \Phi_{k+1} \widehat{\mathbf{B}}_{k}^{\mathrm{uu}} \Phi_{k+1}^{\mathrm{T}}$$

$$+ \mathbf{U}_{k+1}^{\mathrm{fr}} \left[ \mathbf{L} - \widehat{\mathbf{K}}_{k} \mathbf{H}_{k} \mathbf{E}_{k}^{\mathrm{f}} \right] \widehat{\mathbf{B}}_{k}^{\mathrm{fu}} \Phi_{k+1}^{\mathrm{T}}$$

$$(37b)$$

$$(37c)$$

5

$$+ \mathbf{U}_{k+1}^{\text{ff}} \left[ \mathbf{I}_{r} - \mathbf{K}_{k} \mathbf{H}_{k} \mathbf{E}_{k}^{\text{f}} \right] \mathbf{B}_{k}^{\text{fu}} \mathbf{\Phi}_{k+1}^{\text{T}}$$

[revised manuscript text omitted]